

# A systematic interpolatory method for an impurity in a one-dimensional fermionic background

**Erik J. Lindgren[1⋆], Rafael E. Barfknecht[2,3†] and Nikolaj T. Zinner[3‡]**

**1** Nordita, KTH Royal Institute of Technology and Stockholm University,
Roslagstullsbacken 23, SE-106 91 Stockholm, Sweden
**2** Instituto de Física da UFRGS, Av. Bento Gonçalves 9500, Porto Alegre, RS, Brazil
**3** Department of Physics and Astronomy, Aarhus University, Ny Munkegade 120, Denmark

⋆ ejonathanlindgren@gmail.com, † barfknecht@lens.unifi.it, ‡ zinner@aias.au.dk

## Abstract

We explore a new numerical method for studying one-dimensional quantum systems in a trapping potential. We focus on the setup of an impurity in a fermionic background, where a single distinguishable particle interacts through a contact potential with a number of identical fermions. We can accurately describe this system, for various particle numbers, different trapping potentials and arbitrary finite repulsion, by constructing a truncated basis containing states at both zero and infinite repulsion. The results are compared with matrix product states methods and with the analytical result for two particles in a harmonic well.



# 1 Introduction

The investigation of one-dimensional quantum systems of interacting particles has, in the last decades, attracted renewed interest due to striking advances in experiments with cold atoms in optical traps [1]. Paradigmatic models extensively explored in the fields of condensed matter [2–5] and mathematical physics [6–8] are now within reach of experiments, and their exotic properties can be measure with great precision. Moreover, the degree of control over several experimental parameters, including interactions between the atoms [9–12] and trapping geometries opens up the possibility of using such experiments as quantum simulators for a multitude of interesting models [13], even beyond usual condensed matter systems [14].

One particular problem which has attracted interest in this context is that of a single distinct atom (or impurity) embedded in a background of identical particles. In the field of condensed matter, such systems can present interesting phenomena, such as the Kondo effect [15] and the orthogonality catastrophe [16]. Theoretical and experimental studies with ultracold atomic setups have extensively explored both the bosonic [17–27] and fermionic [28–37] manifestations of these models - the so-called Bose and Fermi polarons, respectively. The one-dimensional fermionic case, in particular, dates back to McGuire's impurity model in a homogeneous ge-

ometry [38, 39], which is exactly solvable through the Bethe ansatz approach [40]. Other approaches have later generalized the study of static properties to mixed fermionic systems in harmonic potentials [41–47]. On the dynamical side, impurity models have been shown to present exotic effects such as Bloch oscillations [48] and quantum flutter [49, 50].

In this work we present an original way to obtain the static properties of an impurity in a fermionic background, where the number of background fermions can be arbitrarily modified. We employ a variational principle where our ansatz for the wavefunction is a combination of states at zero and infinite interaction, relying on the fact that the analytical expressions for these limits are known. In practice, we construct a truncated basis by choosing a certain number of states at each limit and then employ the Gram-Schmidt ortonormalization process to construct an orthonormal basis. By diagonalizing the Hamiltonian in this basis, we obtain an approximation for the wavefunctions and eigenvalues. While it may be difficult to reach a regime of strong interactions with usual methods, our approach is exact in the zero and infinite interaction limits. This method is an extension of Ref. [51], where only two basis states were used. It can be applied to systems in different trapping geometries, and the repulsive interactions can be tuned from weak to strong. To validate our method, we compare our results for spatial densities and momentum distributions to simulations of the continuum obtained with Matrix Product States (MPS) [52, 53], as well as the known analytical results for two atoms in a harmonic trap [54].

## 2 Hamiltonian

We focus on a one-dimensional system of $N$ identical fermions (majority) which interacts with a single distinct particle (minority) with the same mass in the presence of a trapping potential. The Hamiltonian can be written as

$$H = \sum_{i=0}^{N} \left( \frac{\mathbf{p}_i^2}{2m} + V(\mathbf{x}_i) \right) + g \sum_{i=1}^{N} V_{0,i}(\mathbf{x}_0 - \mathbf{x}_i), \tag{1}$$

where the potential $V$ is the background potential (which in this paper is either a harmonic potential, $V(x) = m\omega^2 x^2/2$, or the double well constructed in Appendix C with parameters as shown in Figure 1) and $V_{0,i}$ is the interaction between the minority and majority, namely we have $\langle x_0, x_1, \ldots, x_N | V_{0,i} | x_0', x_1', \ldots, x_N' \rangle = v(x_0 - x_i)\delta(x_0 - x_0') \cdots \delta(x_N - x_N')$. In our case of a contact interaction, we have $v(x) = \delta(x)$. Since all particles have the same mass, we can interpret the single impurity as a fermionic atom in a different internal state than the remaining majority atoms. Such systems can be realized in the lab with ultracold Li atoms in different hyperfine states [4].

For simplicity we will work in units where $\hbar = 1$ and $m = 1$, but we will reinstate these units in all figures.

## 3 Variational method

Our variational method consists of constructing a suitable truncated basis of states. The basis states are constructed by using both the analytically known eigenstates at zero interactions as well as the analytically known solutions at infinite interaction.

## 3.1 States at zero interaction

The states at zero interaction are denoted by $|\phi_i\rangle$, for $0 \leq i \leq n-1$. Each state $|\phi_i\rangle$ is defined by a collective index $\vec{k}^{(i)}$ of $N+1$ single particle states, namely

$$\vec{k}^{(i)} = [k_0^{(i)}; k_1^{(i)}, \ldots, k_N^{(i)}]. \tag{2}$$

Note that for the $k_1^{(i)}, \ldots, k_N^{(i)}$, different orders correspond to the same state up to a sign since they correspond to the majority particles, while the single $k_0^{(i)}$ corresponds to the quantum number of the minority particle. We assume that $k_1^{(i)} < \ldots < k_N^{(i)}$. We define the totally antisymmetric state of a number of $M$ (ordered) quantum states $\vec{v}$ by

$$|\Phi_{\vec{v}}\rangle = \frac{1}{\sqrt{M!}} \sum_{\sigma} \text{sign}(\sigma) |f_{v_{\sigma(1)}}\rangle \cdots |f_{v_{\sigma(M)}}\rangle. \tag{3}$$

Let us further denote $\vec{v}[i, j, \ldots]$ as the (ordered) set $\vec{v}$ with $v_i$, $v_j$, …, removed. This notation will be used throughout this article. A zero interaction state is then of the form

$$|\phi_i\rangle = |f_{k_0^{(i)}}\rangle |\Phi_{\vec{k}^{(i)}[0]}\rangle. \tag{4}$$

## 3.2 States at infinite interaction

At infinite interaction, the states are denoted as $|\psi_\mu\rangle$, for $0 \leq \mu \leq m-1$. Note that the number of states at zero interaction, $n$, is not necessarily the same as the number of states at infinite interaction. Each state $|\psi_\mu\rangle$ corresponds to a collective index $\vec{q}^{(\mu)}$ of $N+1$ single particle states corresponding to a completely antisymmetric state $\Phi_{\vec{q}_\mu}$ built from the quantum states

$$\vec{q}^{(\mu)} = [q_0^{(\mu)}, \ldots, q_N^{(\mu)}], \tag{5}$$

as well as a set of $N+1$ coefficients $\vec{a}^{(\mu)}$,

$$\vec{a}^{(\mu)} = [a_0^{(\mu)}, \ldots, a_N^{(\mu)}]. \tag{6}$$

Note that different orders of the $q_i^{(\mu)}$ correspond to the same state up to a sign, and we will assume that $q_0^{(i)} < \ldots < q_N^{(i)}$, and we will assume that the $\vec{a}^{(\mu)}$ satisfy

$$\sum_{l=0}^{N} (a_l^{(\mu)})^2 = N+1, \quad \sum_{l=0}^{N} a_l^{(\mu)} a_l^{(\nu)} = 0, \quad \mu \neq \nu. \tag{7}$$

The state at infinite interaction is then defined in the coordinate representation as

$$\psi_\mu(x_0, \ldots, x_N) = a_l^{(\mu)} \Phi_{\vec{q}^{(\mu)}}(x_0, \ldots, x_N), \qquad \text{when } (x_0, \ldots, x_N) \in \mathcal{M}_l, \tag{8}$$

where we denote $\mathcal{M}_l$ as the set of points where $x_0$, the coordinate for the minority particle, is smaller than exactly $l$ of the $x_1, \ldots, x_N$.

These exact solutions of the Hamiltonian (1) at $g = +\infty$ [43, 44], are orthogonal and properly normalized to unity provided that (7) holds. However, they are not orthogonal to the zero interaction eigenstates, and in Section 3.4 we will apply the Gram-Schmidt process to construct an orthonormal basis.

## 3.3 Overlaps between the zero and infinite interaction states

In this section we will compute the overlaps between the infinite interaction states $\psi_\mu$ and the zero interaction states $\phi_i$, which is a necessary input for the construction in Section 3.4 and for computing the matrix elements and overlaps that include the states $\chi_\mu$. We will denote the overlaps between the states at zero interaction and at infinite interaction by $C_{i\mu}$, where by convention $i \in (0,\ldots,n-1)$ corresponds to the index for the zero interaction states and $\mu \in (0,\ldots,m-1)$ corresponds to the index for the states at infinite interaction. The zero interaction state is on the form

$$\phi_i(x_0,\ldots,x_N) = f_{k_0^{(i)}}(x_0)\Phi_{\vec{k}^{(i)}[0]}(x_1,\ldots,x_N), \tag{9}$$

where $\Phi_{\vec{k}^{(i)}[0]}$ is again the totally antisymmetric wave function

$$\Phi_{\vec{k}^{(i)}[0]} = \frac{1}{\sqrt{N!}}\sum_\pi \mathrm{sign}(\pi) f_{k_{\pi(1)}^{(i)}}\cdots f_{k_{\pi(N)}^{(i)}}. \tag{10}$$

Again, we use the notation where $\vec{k}^{(i)}[J]$ is the set $(k_0^{(i)},\ldots,k_N^{(i)})$ with $k_J^{(i)}$ removed. Recall that at $g = +\infty$, an eigenstate can be specified by a sequence of $N+1$ numbers $a_l$, $0 \le l \le N$, as well as a set $\vec{q}^{(\mu)} = (q_0^{(\mu)},\ldots,q_N^{(\mu)})$ of single particle quantum numbers, and is constructed by

$$\psi_\mu = a_l \Phi_{\vec{q}^{(\mu)}}(x_0,x_1,\ldots,x_N), \; x_1,\ldots,x_N \in \mathcal{M}_l(x_0), \tag{11}$$

where $\mathcal{M}_l(x_0)$ is the set where $x_0$ is smaller than exactly $l$ of the $x_j$ with $j \ge 1$. The overlaps are thus given by

$$C_{i\mu} = \sum_{l=0}^N a_l I_l, \tag{12}$$

where

$$
\begin{aligned}
I_l &\equiv \int_{-\infty}^\infty \mathrm{d}x_0 \int_{\mathcal{M}_l(x_0)} \phi_i(x_0,\ldots,x_N)\psi_\mu(x_0,\ldots,x_N)\mathrm{d}x_1\cdots\mathrm{d}x_N \\
&= \frac{1}{\sqrt{N+1}}\sum_{J=0}^N (-1)^J \int_{-\infty}^\infty \mathrm{d}x_0 f_{k_0^{(i)}}(x_0) f_{q_J^{(\mu)}}(x_0) \int_{\mathcal{M}_l(x_0)} \Phi_{\vec{k}^{(i)}[0]}\Phi_{\vec{q}^{(\mu)}[J]} \\
&= \frac{1}{l!\sqrt{N+1}}\sum_{J=0}^N (-1)^J \int_{-\infty}^\infty \mathrm{d}x_0 f_{k_0^{(i)}}(x_0) f_{q_J^{(\mu)}}(x_0) \partial_\epsilon^l \det(A^J + \epsilon B^J)_{\epsilon=0}, \tag{13}
\end{aligned}
$$

where in the last step we have used the formula in Appendix A.1. The matrix $A^J$ is defined by $A_{kl}^J = \int_{-\infty}^{x_0} f_{k_{k+1}^{(i)}} f_{q_l^{(\mu)}}$ for $l < J$ and $A_{kl}^J = \int_{-\infty}^{x_0} f_{k_{k+1}^{(i)}} f_{q_l^{(\mu)}}$ for $l \ge J$ while $B^J$ is defined by $B_{kl}^J = \int_{x_0}^\infty f_{k_{k+1}^{(i)}} f_{q_l^{(\mu)}}$ for $l < J$ and $B_{kl}^J = \int_{x_0}^\infty f_{k_{k+1}^{(i)}} f_{q_{l+1}^{(\mu)}}$ for $l \ge J$. To compute the derivatives efficiently we evaluate the determinant for several values of $\epsilon$, linearly spaced in $(-1,1)$, and fit a polynomial. We will encounter similar, but more involved, calculations when we compute the densities.

## 3.4 Constructing the basis

We will construct our basis by starting with the $n$ states at zero interaction. We then add the infinite interaction states one by one, and orthonormalize after each added state. In other words, each state at infinite interaction $|\psi_\mu\rangle$ corresponds to a state $|\chi_\mu\rangle$, which is a linear

combination of all the $|\phi_i\rangle$ and the $|\chi_\nu\rangle$ with $\nu < \mu$ such that it is orthogonal to all of these states. This procedure will be explained below.

We define $C_{i\mu} = \langle\phi_i|\psi_\mu\rangle$ and $W_{\mu\nu} = \langle\chi_\mu|\psi_\nu\rangle$. Note that neither of these matrices are symmetric. The states $|\chi_\mu\rangle$ are then given by

$$|\chi_\mu\rangle = N_\mu\left(|\psi_\mu\rangle - \sum_{i=0}^{n-1} C_{i\mu}|\phi_i\rangle - \sum_{\rho=0}^{\mu-1} W_{\rho\mu}|\chi_\rho\rangle\right), \tag{14}$$

where the normalization constant is given by

$$N_\mu = \left(1 - \sum_{i=0}^{n-1} C_{i\mu}^2 - \sum_{\rho=0}^{\mu-1} W_{\rho\mu}^2\right)^{-1/2}. \tag{15}$$

The $W_{\mu\nu}$ can be computed inductively. We first have that

$$W_{0\nu} = N_0\left(\delta_{0\nu} - \sum_{i=0}^{n-1} C_{i0}C_{i\nu}\right). \tag{16}$$

Then, for any $\mu$, assuming knowledge of $W_{\rho\sigma}$ where $\rho \leq \mu - 1$, we can compute $W_{\mu\nu}$ as

$$W_{\mu\nu} = N_\mu\left(\delta_{\mu\nu} - \sum_{i=0}^{n-1} C_{i\mu}C_{i\nu} - \sum_{\rho=0}^{\mu-1} W_{\rho\mu}W_{\rho\nu}\right), \tag{17}$$

where $N_\mu$ is also given in terms of known $W_{\rho\sigma}$. Given the $W_{\mu\nu}$, $C_{i\mu}$ and $N_\mu$ we now know our truncated basis $(|\phi_0\rangle,\ldots,|\phi_{n-1}\rangle,|\chi_0\rangle,\ldots,|\chi_{m-1}\rangle) \equiv (\xi_0,\ldots,\xi_{m+n-1})$. We will then express our Hamiltonian in this basis and numerically diagonalize it to find approximations to the eigenstates and energies.

Note that the construction of the basis $\xi_i$ in this subsection assumes that none of the states are linearly dependent. The only states at infinite interaction that can be linearly dependent with the states at zero interaction are the totally antisymmetric states, namely the states with coefficients $a_0^{(\mu)} = \ldots = a_N^{(\mu)} = 1$, and thus one may have to exclude some of these states (or exclude some states at zero interaction) such that the final basis only contains states that are linearly independent. However, note that totally antisymmetric states are anyway already eigenstates to the Hamiltonian at finite $g$, and thus can be safely removed since they will be orthogonal to any non-trivial eigenstates. We will thus always exclude totally antisymmetric states at infinite interaction.

## 3.5 The Hamiltonian expressed in the basis

We will now express the Hamiltonian in the $|\xi_i\rangle$ basis by computing $\langle\xi_i|H|\xi_j\rangle$. We will write the Hamiltonian as

$$H = H_0 + gV, \tag{18}$$

where $V$ is the contact interaction between the majority and minority particles and $H_0$ is the Hamiltonian at zero interaction. We will treat these two terms individually.

For the zero interaction states, we have $\langle\phi_i|H_0|\phi_j\rangle = \delta_{ij}E_i$, $\langle\phi_i|H_0|\chi_\mu\rangle = 0$ due to the orthogonality propery of the basis and also $\langle\psi_\mu|H_0|\psi_\nu\rangle = \delta_{\mu\nu}E_\mu$. Let us define the quantity

$$L_{\mu\nu} = \langle\psi_\mu|H|\chi_\nu\rangle = N_\nu\left(E_\mu\delta_{\mu\nu} - \sum_{i=0}^{n-1} C_{i\mu}C_{i\nu}E_i - \sum_{\rho=0}^{\nu-1} W_{\rho\nu}L_{\mu\rho}\right). \tag{19}$$

These can be computed recursively, starting with the known $L_{\mu 0}$. The matrix elements $\langle \chi_\mu | H_0 | \chi_\nu \rangle$ are then given by

$$\langle \chi_\mu | H_0 | \chi_\nu \rangle = N_\mu \left( L_{\mu\nu} - \sum_{\rho=0}^{\mu-1} W_{\rho\mu} \langle \chi_\rho | H_0 | \chi_\nu \rangle \right), \tag{20}$$

which can also be calculated recursively starting with the known $\langle \chi_0 | H_0 | \chi_\nu \rangle$.

Now let us look at the interaction operator $V$. Note that $\langle \xi_i | V | \psi_\mu \rangle = 0$ since $V$ is a contact interaction and $\langle x | \psi_\mu \rangle$ vanishes when $x_0 = x_j$ for $1 \le j \le N$. Let us define $V_{ij} = \langle \phi_i | V | \phi_j \rangle$. We then have

$$V_{i\mu} = N_\mu \left( - \sum_{j=0}^{n-1} V_{ij} C_{j\mu} - \sum_{\sigma=0}^{\mu-1} W_{\sigma\mu} V_{i\sigma} \right), \tag{21}$$

which can be computed recursively starting with $V_{i,\mu=0}$. Given these quantities, the remaining matrix elements can be calculated as

$$V_{\mu\nu} = N_\mu \left( - \sum_{j=0}^{n-1} C_{j\mu} V_{j\nu} - \sum_{\sigma=0}^{\mu-1} W_{\sigma\mu} V_{\sigma\nu} \right). \tag{22}$$

To compute the matrix elements $V_{ij}$, note that the interaction operator between two particles, which we denote by $V_2$, is defined as

$$\langle x_1, x_2 | V_2 | y_1, y_2 \rangle = \delta(x_1 - y_1) \delta(y_1 - y_2) \delta(x_2 - y_2), \tag{23}$$

where $\langle x_1, x_2 |$ is a position space eigenstate for two particles. Now consider some discrete set of single particle states $|i\rangle$, $i = 0, 1, \ldots$, defined by the wavefunction by $\langle x | i \rangle = f_i(x)$. The matrix element for the interaction operator between two particles in this basis is then

$$\langle k_1, k_2 | V_2 | k_1', k_2' \rangle = \int_{-\infty}^{\infty} f_{k_1}(x) f_{k_2}(x) f_{k_1'}(x) f_{k_2'}(x). \tag{24}$$

Now we would like to know the matrix elements of the total interaction operator between two many-body states $\vec{k}^{(i)} = [k_0^{(i)}; k_1^{(i)}, \ldots, k_N^{(i)}]$ and $\vec{k}^{(j)} = [k_0^{(j)}; k_1^{(j)}, \ldots, k_N^{(j)}]$ (and we again denote $\vec{k}^{(i)}[0] = [k_1^{(i)}, \ldots, k_N^{(i)}]$, and we assume $k_1^{(i)} > \ldots > k_N^{(i)}$). The total interaction operator is given as $V = \sum_{l=1}^{N} V_{0l}$, where $V_{0l}$ is the interaction operator between particle 0 (the impurity) and particle with index $l$. For two sets $A$ and $B$ with equal size, let us define $|A - B|$ be the number of elements that only appear in $A$ (or equivalently in only $B$). For a particle with index $p$, with $p \ne 0, l$, $V_{0l}$ is diagonal and thus we have

$$\langle \vec{k}^{(i)} | V | \vec{k}^{(j)} \rangle = 0, \tag{25}$$

if $|\vec{k}^{(i)}[0] - \vec{k}^{(j)}[0]| > 1$. If $|\vec{k}^{(i)}[0] - \vec{k}^{(j)}[0]| = 1$, we obtain

$$\langle \vec{k}^{(i)} | V | \vec{k}^{(j)} \rangle = \langle k_0^{(i)}, k_l^{(i)} | V_2 | k_0^{(j)}, k_{l'}^{(j)} \rangle (-1)^{l-l'}, \tag{26}$$

where $l$ and $l'$ are the unique indices such that $k_l^{(i)} \ne k_{l'}^{(j)}$ and $\vec{k}^{(i)}[0, l] = \vec{k}^{(j)}[0, l']$. If $\vec{k}^{(i)}[0] = \vec{k}^{(j)}[0]$, we instead obtain

$$\langle \vec{k}^{(i)} | V | \vec{k}^{(j)} \rangle = \sum_{l=1}^{N} \langle k_0^{(i)}, k_l^{(i)} | V_2 | k_0^{(j)}, k_l^{(j)} \rangle. \tag{27}$$

This concludes our construction of the Hamiltonian $H = H_0 + gV$, and all that remains is diagonalizing the matrix $\langle \xi_i | H | \xi_j \rangle$ to find the energies and wavefunctions.

# 4 Observables

In this section we explain how to compute several important observables. They will all be computed starting with a specific eigenstate, which we denote by $|\Psi\rangle$, or $\Psi(x_0,\ldots,x_N) = \langle x_0,\ldots,x_N|\Psi\rangle$ in the coordinate basis. This state is expressed as a linear combination of the zero interaction states and infinite interaction states

$$|\Psi\rangle = \sum_{i=0}^{n-1} C_i|\phi_i\rangle + \sum_{\mu=0}^{m-1} D_\mu|\psi_\mu\rangle, \tag{28}$$

which can be obtained easily given the expansion of $\Psi$ in the basis $\{\phi_i, \chi_\mu\}$. Note that the $\{\phi_i, \psi_\mu\}$ is not an orthonormal basis.

We will start by computing the single particle density, which is the easiest observable presented in this section. The equations for the other observables are similar in nature but with varying extra degrees of complexity and subtleties, and thus it is recommended to understand the single particle density computation in detail first.

## 4.1 Single particle minority density matrix

The single particle density matrix is defined by integrating out the coordinates of the majority particles as

$$
\begin{aligned}
\rho(x_0, y_0) &= \int \Psi^*(x_0, x_1, \ldots, x_N)\Psi(y_0, x_1, \ldots, x_N) \\
&= \int \sum_{i=0,j=0}^{n-1} C_i^* C_j \phi_i^*(x_0, x_1, \ldots, x_N)\phi_j(y_0, x_1, \ldots, x_N) + \\
&+ \int \sum_{i=0}^{n-1}\sum_{\mu=0}^{m-1} \Big( C_i^* D_\mu \phi_i^*(x_0, x_1, \ldots, x_N)\psi_\mu(y_0, x_1, \ldots, x_N) + \\
&+ C_i D_\mu^* \phi_i(y_0, x_1, \ldots, x_N)\psi_\mu^*(x_0, x_1, \ldots, x_N) \Big) + \\
&+ \int \sum_{\mu=0,\nu=0}^{m-1} D_\mu^* D_\nu \psi_\mu^*(x)\psi_\nu(x) \\
&\equiv \sum_{i,j} C_i^* C_j \alpha_{i,j}(x_0, y_0) + \sum_{i,\mu}\Big( C_i^* D_\mu \beta_{i,\mu}(x_0, y_0) + \\
&+ C_i D_\mu^* \beta_{i,\mu}^*(y_0, x_0) \Big) + \sum_{\mu,\nu} D_\mu^* D_\nu \gamma_{\mu,\nu}(x_0, y_0),
\end{aligned}
\tag{29}
$$

where the integral is short for $\int = \int_{-\infty}^{\infty} dx_1 \cdots \int_{-\infty}^{\infty} dx_N$. The density matrix is useful since it is related to the momentum distribution by a simple Fourier transform. For just the particle density in coordinate space, we set $x_0 = y_0$. We will comment on how the computations simplify for this special case.

The simplest term, namely between the zero interaction states is given by

$$\alpha_{i,j}(x_0, y_0) = f_{k_0^{(i)}}^*(x_0) f_{k_0^{(j)}}(y_0)\delta_{\vec{k}^{(i)}[0], \vec{k}^{(j)}[0]}. \tag{30}$$

Here we are again using the notation that $\vec{k}[0]$ is equal to $\vec{k}$ with $k_0$ removed, namely the set $\{k_1, \ldots, k_N\}$, and the Kronecker delta is thus equal to one if and only if the sets $\{k_1^{(i)}, \ldots, k_N^{(i)}\}$

and $\{k_1^{(j)}, \ldots, k_N^{(j)}\}$ are the same.

For the cross terms $\beta_{i,\mu}$, it will be useful to split up the integral into several regions, and we write

$$\int = \sum_{l=0}^{N} \int_{\mathcal{M}_l}, \tag{31}$$

where $\mathcal{M}_l$ as the set of points where $y_0$ is smaller than exactly $l$ of the $x_1, \ldots, x_N$. We then split up the term $\beta_{i,\mu}(x_0, y_0)$ as

$$\beta_{i,\mu}(x_0, y_0) = \sum_l \beta_{i,\mu}^l(x_0, y_0). \tag{32}$$

The cross term is then given by

$$\beta_{i,\mu}^l(x_0, y_0) = \frac{a_l}{\sqrt{N+1}} \sum_{J=0}^{N} (-1)^J f_{k_0^{(i)}}(x_0) f_{q_J^{(\mu)}}(y_0) \int_{\mathcal{M}_l} \Phi_{\vec{k}^{(i)}[0]}(x_1, \ldots, x_N) \Phi_{\vec{q}^{(\mu)}[J]}(x_1, \ldots, x_N)$$

$$= \frac{a_l}{l! \sqrt{N+1}} \sum_{J=0}^{N} (-1)^J f_{k_0^{(i)}}(x_0) f_{q_J^{(\mu)}}(y_0) \partial_\epsilon^l \det(A^J + \epsilon B^J)_{\epsilon=0}(y_0), \tag{33}$$

where $\Phi$ represents a totally antisymmetric state. The matrix $A^J$ is defined by $A_{ab}^J = \int_{-\infty}^{y_0} f_{k_{a+1}^{(i)}} f_{q_b^{(\mu)}}$ for $b < J$ and $A_{ab}^J = \int_{-\infty}^{y_0} f_{k_{a+1}^{(i)}} f_{q_{b+1}^{(\mu)}}$ for $b \geq J$ while $B^J$ is defined by $B_{ab}^J = \int_{y_0}^{\infty} f_{k_{a+1}^{(i)}} f_{q_b^{(\mu)}}$ for $b < J$ and $B_{ab}^J = \int_{y_0}^{\infty} f_{k_{a+1}^{(i)}} f_{q_{b+1}^{(\mu)}}$ for $b \geq J$. Here $a$ and $b$ take the values $0, \ldots, N-1$. For a derivation of this equation see A.1.

For the density where $x_0 = y_0$, this works also for the infinite interaction terms, namely we can write

$$\gamma_{\mu,\nu}^l(x_0) = \frac{a_l^2}{N+1} \sum_{J,J'=0}^{N} (-1)^{J+J'} f_{q_J^{(\mu)}}(x_0) f_{q_{J'}^{(\nu)}}(x_0) \int_{\mathcal{M}_l} \Phi_{\vec{q}^{(\mu)}[J]} \Phi_{\vec{q}^{(\nu)}[J']}$$

$$= \frac{a_l^2}{l!(N+1)} \sum_{J,J'=0}^{N} (-1)^{J+J'} f_{q_J^{(\mu)}}(x_0) f_{q_{J'}^{(\nu)}}(x_0) \partial_\epsilon^l \det(A^{J,J'} + \epsilon B^{J,J'})_{\epsilon=0}(x_0), \tag{34}$$

where now the matrices $A^{J,J'}$ and $B^{J,J'}$ are defined by $A_{ab}^{J,J'} = \int_{-\infty}^{x_0} f_{q_{a+\sigma}^{(\mu)}} f_{q_{b+\delta}^{(\nu)}}$ and $B_{ab}^{J,J'} = \int_{x_0}^{\infty} f_{q_{a+\sigma}^{(\mu)}} f_{q_{b+\delta}^{(\nu)}}$ where $\sigma = 0$ for $a < J$, $\sigma = 1$ for $a \geq J$, $\delta = 0$ for $b < J'$ and $\delta = 1$ for $b \geq J'$. Here $a$ and $b$ take the values $0, \ldots, N-1$ and we refer again to A.1 for a derivation of the determinant formulas.

However, when $x_0 \neq y_0$, it is necessary to split the integral in more regions. We then write

$$\int = \sum_{l=0, s=0}^{N} \int_{\mathcal{M}_{l,s}}, \tag{35}$$

where $\mathcal{M}_{l,s}$ is the region where $x_0$ and $y_0$ are smaller than exactly $l$ respectively $s$ of the $x_1, \ldots, x_N$. We then split up the terms $\gamma_{\mu,\nu}(x_0, y_0)$ as

$$\gamma_{\mu,\nu}(x_0, y_0) = \sum_{l,s} \gamma_{\mu,\nu}^{l,s}(x_0, y_0). \tag{36}$$

The term only involving infinite interaction states is then given by

$$\gamma_{\mu,\nu}^{l,s}(x_0, y_0) = \frac{a_l a_s}{N+1} \sum_{J,J'=0}^{N} (-1)^{J+J'} f_{q_J^{(\mu)}}(x_0) f_{q_{J'}^{(\nu)}}(y_0) \int_{\mathcal{M}_{l,s}} \Phi_{\vec{q}^{(\mu)}[J]} \Phi_{\vec{q}^{(\nu)}[J']}$$

$$= \frac{a_l a_s}{|l-s|!(\min(l,s))! \sqrt{N+1}} \sum_{J,J'=0}^{N} (-1)^{J+J'} f_{q_J^{(\mu)}}(x_0) f_{q_{J'}^{(\nu)}}(y_0) \times$$

$$\partial_\epsilon^{|l-s|} \partial_\tau^{\min(l,s)} \det(A^{J,J'} + \epsilon B^{J,J'} + \tau C^{J,J'})(x_0, y_0)|_{\epsilon=0, \tau=0}. \tag{37}$$

Now the matrices are defined as $A_{ab} = \int_{-\infty}^{\min(x,x')} f_{q_{a+\sigma}^{(\mu)}}(x'') f_{q_{b+\delta}^{(\nu)}}(x'') dx''$, $B_{ab} = \int_{\min(x,x')}^{\max(x,x')} f_{q_{a+\sigma}^{(\mu)}}(x'') f_{q_{b+\delta}^{(\nu)}}(x'') dx''$ and $C_{ab} = \int_{\max(x,x')}^{\infty} f_{q_{a+\sigma}^{(\mu)}}(x'') f_{q_{b+\delta}^{(\nu)}}(x'') dx''$ where $\sigma = 0$ for $a < J$, $\sigma = 1$ for $a \geq J$, $\delta = 0$ for $b < J'$ and $\delta = 1$ for $b \geq J'$. The indices $a$ and $b$ take the values $0, \ldots, N-1$. See A.2 for a derivation of this formula.

## 4.2 Majority particle density matrix

The single particle majority density matrix is defined by integrating out the coordinate of the single minority particle and the coordinates of $N-1$ of the majority particles. We thus write

$$\rho^{\text{maj}}(x_0, y_0) = \int \Psi^*(x_0, x_1, \ldots, x_N) \Psi(x_0, y_1, \ldots, x_N)$$

$$= \int \sum_{i=0, j=0}^{n-1} C_i^* C_j \phi_i^*(x_0, x_1, \ldots, x_N) \phi_j(x_0, y_1, \ldots, x_N) +$$

$$+ \int \sum_{i=0}^{n-1} \sum_{\mu=0}^{m-1} \left( C_i^* D_\mu \phi_i^*(x_0, x_1, \ldots, x_N) \psi_\mu(x_0, y_1, \ldots, x_N) + \right.$$

$$\left. + C_i D_\mu^* \phi_i(x_0, y_1, \ldots, x_N) \psi_\mu^*(x_0, x_1, \ldots, x_N) \right) +$$

$$+ \int \sum_{\mu=0, \nu=0}^{m-1} D_\mu^* D_\nu \psi_\mu^*(x_0, x_1, \ldots, x_N) \psi_\nu(x_0, y_1, \ldots, x_N)$$

$$\equiv \sum_{i,j} C_i^* C_j \alpha_{i,j}^{\text{maj}}(x_1, y_1) + \sum_{i,\mu} C_i^* D_\mu \int dx_0 \beta_{i,\mu}^{\text{maj}}(x_0, x_1, y_1) +$$

$$+ C_i D_\mu^* \int dx_0 \beta_{i,\mu}^{\text{maj}*}(x_0, x_1, y_1) + \sum_{\mu,\nu} D_\mu^* D_\nu \int dx_0 \gamma_{\mu,\nu}^{\text{maj}}(x_0, x_1, y_1), \tag{38}$$

where in all but the last line the integral is short for $\int = \int_{-\infty}^{\infty} dx_2 \cdots \int_{-\infty}^{\infty} dx_N$ and in the last line we have separated out the $dx_0$ integral in all but the first term. In this case there are not many simplifications when $x_1 = y_1$. The zero interaction term is given by

$$\alpha_{i,j}^{\text{maj}}(x_1, y_1) = \frac{\delta_{k_0^{(i)}, k_0^{(j)}}}{N} \sum_{I=1, J=1}^{N+1} (-1)^{I+J} f_{k_I^{(i)}}^*(x_1) f_{k_J^{(j)}}(y_1) \delta_{\vec{k}^{(i)}[0,I], \vec{k}^{(j)}[0,J]}. \tag{39}$$

The latter delta function means that this expression is zero unless the set $\vec{k}^{(i)}$ with $k_0^{(i)}$ and $k_I^{(i)}$ removed and the set $\vec{k}^{(j)}$ with $k_0^{(j)}$ and $k_J^{(j)}$ removed, are equal, in which case it is equal to one. For the cross term $\beta_{i,\mu}^{\text{maj}}(x_1, y_1)$, we split it up into $N$ terms $\beta^{\text{maj},l}$, corresponding to $x_0$ being

smaller than exactly $l$ of the $x_2, \ldots, x_N$. We have

$$
\begin{aligned}
\beta_{i,\mu}^{\mathrm{maj},l}(x_0, x_1, y_1) = {} & \frac{a^{(\mu)}(l, x_0, y_1)}{\sqrt{N}(N+1)} \sum_{I=1, J=0, J'=0, J' \neq J}^{N+1} sgn(J, J')(-1)^{I+1} f_{k_I^{(i)}}(x_1) f_{q_{J'}^{(\mu)}}(y_1) \\
& f_{k_0^{(i)}}(x_0) f_{q_J^{(\mu)}}(x_0) \int_{\mathcal{M}_l} \Phi_{\vec{k}^{(i)}[0,I]}(x_2, \ldots, x_N) \Phi_{\vec{q}^{(\mu)}[J, J']}(x_2, \ldots, x_N) \\
= {} & \frac{a(l, x_0, y_1)}{l! \sqrt{N}(N+1)} \sum_{I=1, J=0, J'=0, J' \neq J}^{N+1} sgn(J, J')(-1)^{I+1} f_{k_I^{(i)}}(x_1) f_{q_{J'}^{(\mu)}}(y_1) \\
& f_{k_0^{(i)}}(x_0) f_{q_J^{(\mu)}}(x_0) \partial_\epsilon^l \det(A^{0,I;J,J'} + \epsilon B^{0,I;J,J'})_{\epsilon=0}(x_0),
\end{aligned}
\tag{40}
$$

where the integral is again short for $\int = \int_{-\infty}^{\infty} dx_0 dx_2 \cdots \int_{-\infty}^{\infty} dx_N$. The sign $sgn(J, J')$ is defined as $(-1)^{J+J'}$ if $J' < J$ and $-(-1)^{J+J'}$ otherwise. We have defined $a^{(\mu)}(l, x_0, y_1)$ as being equal to $a_l^{(\mu)}$ if $x_0 < y_1$ and equal to $a_{l+1}^{(\mu)}$ otherwise. This formula does not simplify much for the density where $x_1 = y_1$. We have defined the matrices $A_{ab}^{0,I;J,J'} = \int_{-\infty}^{x_0} f_{[\vec{k}^{(i)}(0)(I)]_a}(x') f_{[\vec{q}^{(\mu)}(J)(J')]_b}(x') dx'$ and $B_{ab}^{0,I;J,J'} = \int_{x_0}^{\infty} f_{[\vec{k}^{(i)}(0)(I)]_a}(x') f_{[\vec{q}^{(\mu)}(J)(J')]_b}(x') dx'$, and simplified the notation by assuming that $\vec{S}(I)$ is the (ordered) set $S$ with the element with index $I$ removed.

Now let's consider the term $\gamma$. We now have the expression

$$
\begin{aligned}
\gamma_{\mu,\nu}^{\mathrm{maj},l}(x_0, x_1, y_1) = {} & \frac{a^{(\mu)}(l, x_0, x_1) a^{(\nu)}(l, x_0, y_1)}{N(N+1)} \sum_{I \neq I', J \neq J'}^{N+1} sgn(I, I') sgn(J, J') f_{q_{I'}^{(\mu)}}(x_1) f_{q_{J'}^{(\nu)}}(y_1) \\
& f_{q_I^{(\mu)}}(x_0) f_{q_J^{(\nu)}}(x_0) \int_{\mathcal{M}_l} \Phi_{\vec{q}^{(\mu)}[I, I']}(x_2, \ldots, x_N) \Phi_{\vec{q}^{(\nu)}[J, J']}(x_2, \ldots, x_N) \\
= {} & \frac{a^{(\mu)}(l, x_0, x_1) a^{(\nu)}(l, x_0, y_1)}{l! N(N+1)} \sum_{I \neq I', J \neq J'}^{N+1} sgn(I, I') sgn(J, J') f_{q_{I'}^{(\mu)}}(x_1) f_{q_{J'}^{(\nu)}}(y_1) \\
& f_{q_I^{(\mu)}}(x_0) f_{q_J^{(\nu)}}(x_0) \partial_\epsilon^l \det(A^{I,I';J,J'} + \epsilon B^{I,I';J,J'})_{\epsilon=0}(x_0).
\end{aligned}
\tag{41}
$$

The matrices are now analogously defined, namely

$$
A_{ab}^{I,I';J,J'} = \int_{-\infty}^{x_0} f_{[\vec{q}^{(\mu)}(I)(I')]_a}(x') f_{[\vec{q}^{(\nu)}(J)(J')]_b}(x') dx'
$$

and

$$
B_{ab}^{I,I';J,J'} = \int_{x_0}^{\infty} f_{[\vec{q}^{(\mu)}(I)(I')]_a}(x') f_{[\vec{q}^{(\nu)}(J)(J')]_b}(x') dx'.
$$

### 4.3 Momentum distributions

The momentum distributions are obtained as a Fourier transform of the single particle density matrices. Let us denote the single particle density matrices by $\rho_{\mathrm{min}}(x, y)$ and $\rho_{\mathrm{maj}}(x, y)$ for the minority respectively majority species. The momentum distributions are then defined as

$$
\begin{aligned}
\rho_{\mathrm{min}}(p) &= \frac{1}{2\pi} \int_{-\infty}^{\infty} \int_{-\infty}^{\infty} dx \, dy \, e^{ip(x-y)} \rho_{\mathrm{min}}(x, y) \\
\rho_{\mathrm{maj}}(p) &= \frac{1}{2\pi} \int_{-\infty}^{\infty} \int_{-\infty}^{\infty} dx \, dy \, e^{ip(x-y)} \rho_{\mathrm{maj}}(x, y),
\end{aligned}
\tag{42}
$$

and have the same normalization as the coordinate space densities.

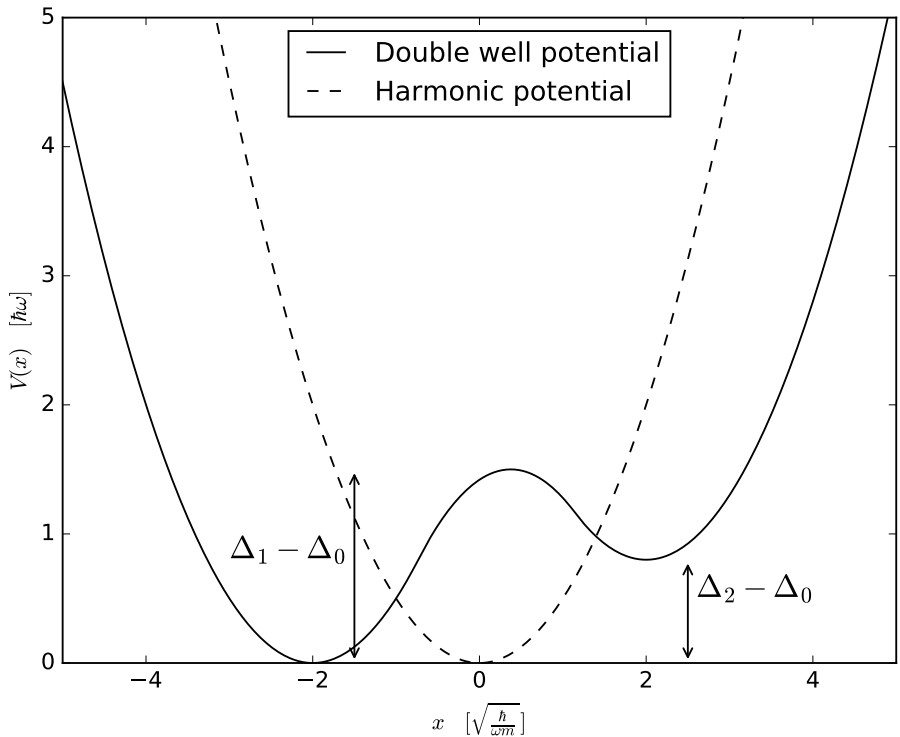

Figure 1: The two potentials used in this paper, the harmonic well with angular frequency $\omega$ and the double well described in Appendix C with parameters $x_2 = -x_0 = 2\sqrt{\frac{\hbar}{\omega m}}$, $\omega_0 = \omega_1 \equiv \omega$, $\Delta_0 = 0$, $\Delta_1 = 1.5 \times \hbar\omega$ and $\Delta_2 = 0.8 \times \hbar\omega$.

## 4.4 Minority-Majority correlation function

The last observable we will consider is the coordinate space minority-majority correlation function, which is defined as

$$\rho(x_0, x_1) = \int \Psi^*(x_0, x_1, \ldots, x_N)\Psi(x_0, x_1, \ldots, x_N), \tag{43}$$

where the integral is short for $\int = \int_{-\infty}^{\infty} dx_2 \cdots \int_{-\infty}^{\infty} dx_N$. The computation of the minority-majority correlation functions is very similar to the computation of the majority density; take the formulas for the majority density matrix and set $x_1 = y_1$, and drop the integrals over $x_0$. Dropping the $x_0$ integral in the $\beta$ and $\gamma$ terms is trivial, and the $\alpha$ term is given by

$$\alpha_{i,j}^{\text{corr}}(x_0, x_1) = \frac{1}{N} f_{k_0^{(i)}}^*(x_0) f_{k_0^{(j)}}(x_0) \sum_{I=1,J=1}^{N+1} (-1)^{I+J} f_{k_I^{(i)}}^*(x_1) f_{k_J^{(j)}}(x_1) \delta_{\vec{k}^{(i)}[0,I],\vec{k}^{(j)}[0,J]}. \tag{44}$$

## 5 Examples

In this section we will consider several examples with different number of particles and in different background potentials. For two particles, we will compare with the analytic result for a harmonic trap. For more particles, we will compare with results computed using matrix

product states as explained in Appendix D.

Recall from Section 3, that when describing the basis we will write $[k_0^{(i)}; k_1^{(i)}, \ldots, k_N^{(i)}]$ to denote a zero interaction state with index $i$, with the single particle in state $k_0^{(i)}$ and the majority particles in the antisymmetric state with quantum numbers $k_1^{(i)} > \ldots > k_N^{(i)}$. A state with index $\mu$ at infinite interaction will be denoted by two sets of numbers, $[q_0^{(\mu)}, \ldots, q_N^{(\mu)}]_\infty$ and $[a_0^{(\mu)}, \ldots, a_N^{(\mu)}]$, such that the wavefunction is a totally antisymmetric wavefunction built from $q_0^{(\mu)} > \ldots > q_N^{(\mu)}$ and which is multiplied with the coefficient $a_l^{(\mu)}$ if $x_0$ is smaller than exactly $l$ of $x_1, \ldots, x_N$.

When truncating the infinite basis (of both zero and infinite interaction states), some choice must be made on how this truncation should be done. In this paper, we will simply truncate when the number of particle excitations above the lowest state is above a certain threshold, namely we will construct the basis from all zero interaction states that satisfy $\sum_a k_a^{(j)} - \sum_a k_a^{(0)} \le \epsilon$ and all infinite interaction states that satisfy $\sum_a q_a^{(\mu)} - \sum_a q_a^{(0)} \le \epsilon$. We will refer to this cutoff as simply the *basis threshold* and denote it by $\epsilon$ in this section. For the harmonic oscillator potential, this is equivalent to an energy cutoff on the states, since the energy is linear in the quantum numbers, but for other potentials this is not the case. We will leave a deeper investigation on the optimal selection of states for future work.

Note that in this paper we work in absolute coordinates and thus our energy spectra, such as figures 2 and 7, show all energy levels. This must be taken into account when comparing with for instance [51], where only the internal energy levels are shown and thus does not include excitations of the center of mass of the particles.

## 5.1 Two particles in a harmonic potential

In this section we will make detailed comparisons between the methods in this paper and the analytically known formula for two particles. A full derivation of the two particle system can be found in Appendix B. The basis is built from states at zero interaction and from states at infinite interaction. The states at zero interaction are specified by two quantum numbers, denoted $[k_0; k_1]$. There are no constraints on these two quantum numbers as we are dealing with two distinguishable particles. At infinite interaction, the states are built by taking a totally antisymmetric state, denoted by $\vec{q} = [q_0, q_1]_\infty$ with $q_0 > q_1$, but by multiplying with different coefficients $a_0$ and $a_1$ depending on the position space coordinates. In other words, the wave function is given by $a_0 \Phi_{\vec{q}}(x_0, x_1)$ when $x_0 > x_1$ and $a_1 \Phi_{\vec{q}}(x_0, x_1)$ when $x_0 < x_1$ where $\Phi$ is the totally antisymmetric state. A basis for such states is given by all antisymmetric states and the coefficients $\vec{a}^{(1)} = [1, 1]$ and $\vec{a}^{(2)} = [1, -1]$. However, note that $\vec{a} = [1, 1]$ just corresponds to the totally antisymmetric state and is thus included among (a linear combination of) the zero interaction states. As was mentioned in the end of Section 3.4, it is important to exclude such linearly dependent states to avoid singular behaviour in the Gram-Schmidt orthogonalization process when constructing the basis. We will thus exclude the states with coefficients $\vec{a}^{(1)} = [1, 1]$ and thus for two particles it is enough to specify a state at infinite interaction only by the quantum numbers $\vec{q} = [q_0, q_1]$ and we leave the coefficients $[1, -1]$ implicit. For example, for a basis threshold of $\epsilon = 2$, we have the zero interaction states $[0, 0], [0, 1], [1, 0], [2, 0], [0, 2]$ and $[1, 1]$, as well as the infinite interaction states $[0, 1]_\infty, [0, 2]_\infty, [0, 3]_\infty$ and $[1, 2]_\infty$ (with the implicit coefficients $[1, -1]$).

### 5.1.1 Energies

In Figure 2 we show the energy of the lowest six states computed using our variational approach and the analytic formula, for various values of the coupling $g$. The ground state interpolates between the state $[0,0]$ at $g = 0$ to the state $[1,0]_\infty$ at $g = +\infty$ (with the implicit coefficients $[1,-1]$).The first and third excited states are totally antisymmetric states (unaffected by the interaction) with the quantum numbers $[0,1]$ and $[0,2]$.

As we can see from this plot, there is an agreement between the results, but it is difficult to appreciate exactly how well they agree. In Figure 3 we therefore plot the energy difference of the analytic result and the variational result for $g = 1/2, 1, 2$ for the ground state and one of the excited states for various basis sizes. As we can see, they agree to an extraordinary accuracy (note the logarithmic scale). Each data point corresponds to a basis constructed with basis threshold $\epsilon = 0, 2, 4, \dots$ The $x$-axis then shows the total size of the basis.

The reason why we look at the fourth excited state is that this is the first "nontrivial" excited state when computed using the analytical formula. As explained in Appendix B, the non-trivial part of the analytical derivation is computing the eigenstates of the relative motion Hamiltonian, and to get the full spectrum we also need to add the energy for the center of mass Hamiltonian which is just a free harmonic oscillator. In the variational method, where we work directly in absolute coordinates, we automatically get all states. It turns out that the first excited state is just a totally antisymmetric state, the second excited state is just the first state plus a center of mass excitation, and the third excited state is then also just a totally antisymmetric state (actually the first excited state plus center of mass motion). The fourth excited state is then the first excited state which corresponds to a non-trivial eigenstate to the relative motion Hamiltonian and where the center of mass energy is zero.

### 5.1.2 Position space densities

In Figure 4 we show the position space density for the ground state at $g = 1$ compared to the analytical result. We see that when we only use two states in the basis there is a small discrepancy between the two methods, but when we use larger basis sizes the methods agree very well. A more detailed comparison can be seen in Figure 5, where we plot the density at three arbitrary values of $x$ as a function of the basis size. We see that they agree exceptionally well with the analytical result. To compute the density from the analytical result, we need to perform an integral transforming from Jacobi coordinates to absolute coordinates (see Appendix B).

### 5.1.3 Momentum space densities

Finally we will compare the momentum densities, which is computed from the density matrix by equation 42. The comparison between the analytical and the variational methods is shown in Figure 6. Again, there is a discrepancy with the analytical result when only using 2 basis states, but when using 10 basis states the results agree very well.

## 5.2 Three particles

In this section we will consider the case of three particles, namely one single particle and two identical fermions. The states at zero interaction are now given by three quantum numbers $[k_0^{(i)}; k_1^{(i)}, k_2^{(i)}]$, and an infinite interaction state is defined using three quantum numbers $[q_0^{(\mu)}, q_1^{(\mu)}, q_2^{(\mu)}]$ and three coefficients $a_l^{(i)}$. The coefficients must be linearly independent and

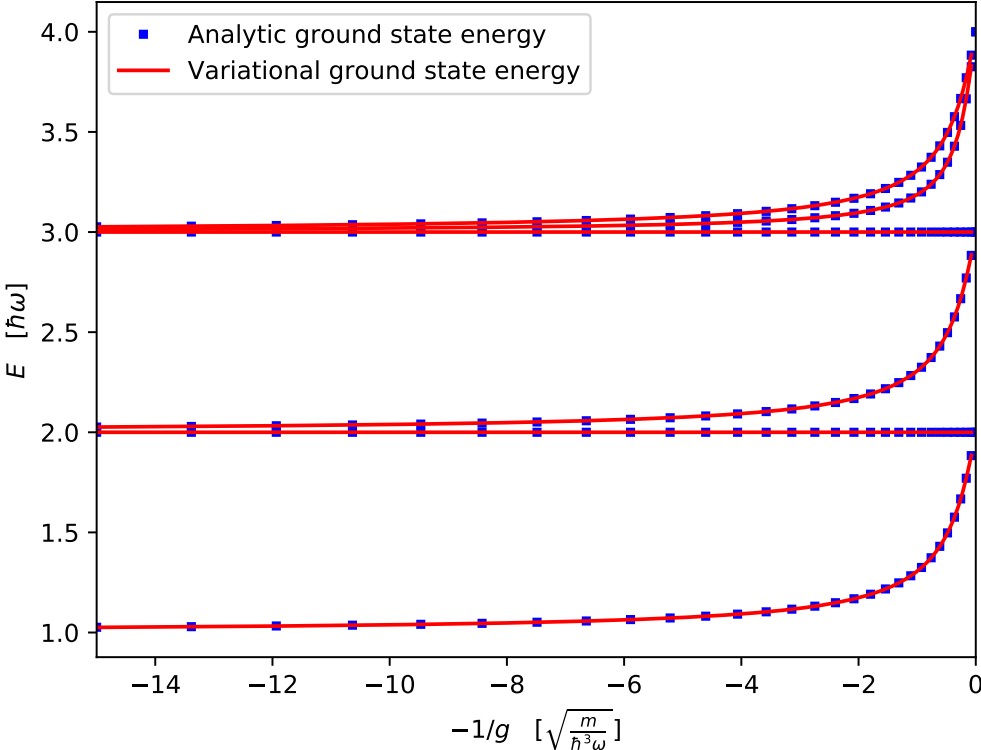

Figure 2: Energies for the lowest six states for the 1+1 system in a harmonic trap, computed both using the exact analytical method and the variational method.

orthogonal, and we will choose the coefficients $[1, 1, 1]$, $[\sqrt{3/2}, 0, \sqrt{3/2}]$ and $[-1/\sqrt{2}, \sqrt{2}, -1/\sqrt{2}]$. For example, for a basis threshold $\epsilon = 2$, we have the 7 zero interaction states $[0; 0, 1]$, $[0; 0, 2]$, $[1; 0, 1]$, $[0; 0, 3]$, $[0; 1, 2]$ and $[1; 0, 2]$ and $[2; 0, 1]$. The infinite interaction states are built from the 4 antisymmetric states $[0, 1, 2]_\infty$, $[0, 1, 3]_\infty$, $[0, 1, 4]_\infty$ and $[0, 2, 3]_\infty$, all multiplied with the corresponding coefficients. Note however, as explained in 3.4, the antisymmetric state $[0, 1, 2]_\infty$ is linearly dependent on some of the zero interaction states, and for simplicity we will thus exclude all antisymmetric states (the states at infinite interaction with coefficients $[1, 1, 1]$).

### 5.2.1 Energies

Figure 7 shows the lowest seven energies for the 2+1 system with harmonic potential. We compare with the matrix product states (MPS) result at $g = 1.0$ for the ground state. Note that to obtain good agreement, we need to compute the energy for several numerical accuracies and then extrapolate the result. The MPS computations for the different accuracies are given by the red dots, and the extrapolated value is the black cross. The dashed lines are the ground state computed with the variational method using basis states with a basis threshold of $\epsilon = 0$, 2, 4 and 6, and the solid lines are computed using a basis cutoff of $\epsilon = 8$. These correspond to basis sizes of 1+2,7+8, 22+22, 50+46 and 95+82 respectively, where the first (second) number is the number of zero (infinite) interaction states the basis is constructed from. Our vatiational method easily gives us the energies of several states at many different values of $g$, which is one of the main advantages of the method compared to for example the MPS method where each computation only yields the energy and wavefunction at one particular interaction.

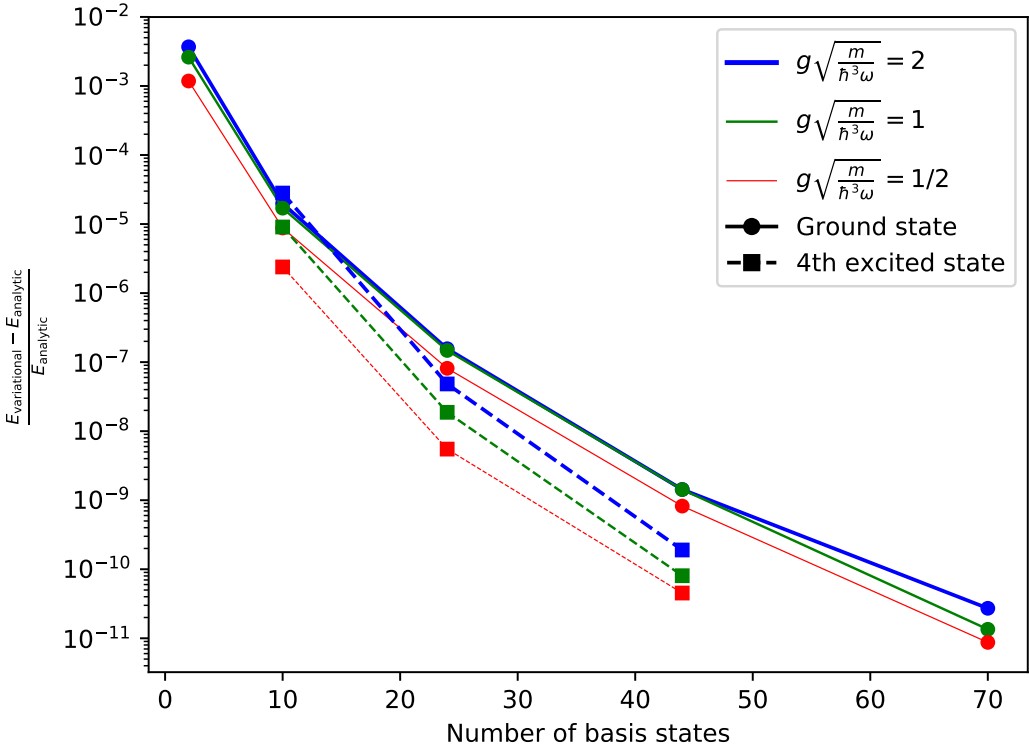

Figure 3: Convergence of the energies in the 1+1 system in a harmonic trap, comparing the variational method to the analytical result.

### 5.2.2 Position space densities

Figure 8 and 9 shows the position space minority and majority density at $g\sqrt{\frac{m}{\hbar^3\omega}}=1$ for the 2+1 system for different basis sizes compared with MPS method. In Figure 8 we assume a harmonic potential, while in Figure 9 we consider the double-well geometry shown in Figure 1. In both cases we have good agreement with the MPS result. The computations are for energy cutoffs of 0, 2 and 4.

In Figure 10 we plot the integral of the squared difference of the densities for different basis sizes to better compare the convergence.

### 5.2.3 Momentum space densities

In Figure 11 we compare the momentum space densities at $g\sqrt{\frac{m}{\hbar^3\omega}}=1$ in the harmonic well with the MPS result. We see that they agree quite well already for the lowest possible number of basis states, and we again see that the discrepeancy goes to zero as we increase the basis size.

## 5.3 Seven particles

In this section we study the 6+1 system. Figure 12 and Figure 13 shows the position space density profiles for the ground state in the double well potential for $g\sqrt{\frac{m}{\hbar^3\omega}}=1$ and $g\sqrt{\frac{m}{\hbar^3\omega}}=10$. We see that for large number of majority particles the system starts to look like a single impu-

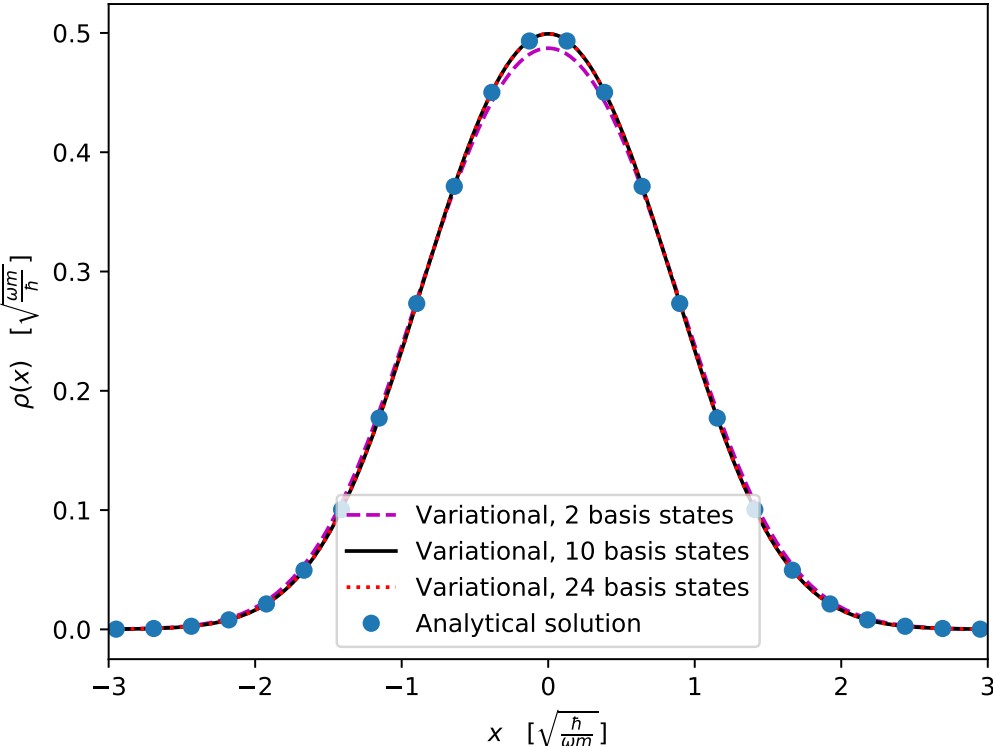

Figure 4: Position space density for the ground state for one of the particles for different basis sizes compared to the analytical result for the 1+1 system in a harmonic well at $g\sqrt{\frac{m}{\hbar^3\omega}}=1$.

rity in a homogeneous bath. Moreover, when the interaction increases the minority particle density clearly gets deformed, which is reproduced with both methods, and we see that our method does work well both for intermediate and strong interactions. However, the discrepancy with the MPS result is clearly larger compared to the 2+1 system.

## 5.4 Eleven particles

Figure 14 shows the position space density profiles for the ground state in the harmonic trap for $g\sqrt{\frac{m}{\hbar^3\omega}}=1$ for the 10+1 system, and we see that the method still works well even for higher particle numbers.

## 6 Conclusions

In this paper we explored a new method for studying strongly coupled one-dimensional systems where an impurity interacts with a background of identical fermions, a method that generalizes that of [51]. Our results compare well both with analytical methods for two particles and with numerical methods based on matrix product states. However, our method has the fundamental advantage of allowing calculations for arbitrary values of the interaction strength by only constructing the basis once. Generally, numerical approaches would require a full calculation for every value of the interaction strength. To compute the eigenstates and energies, we just need to change the interaction parameter $g$ in the Hamiltonian before diagonaliz-

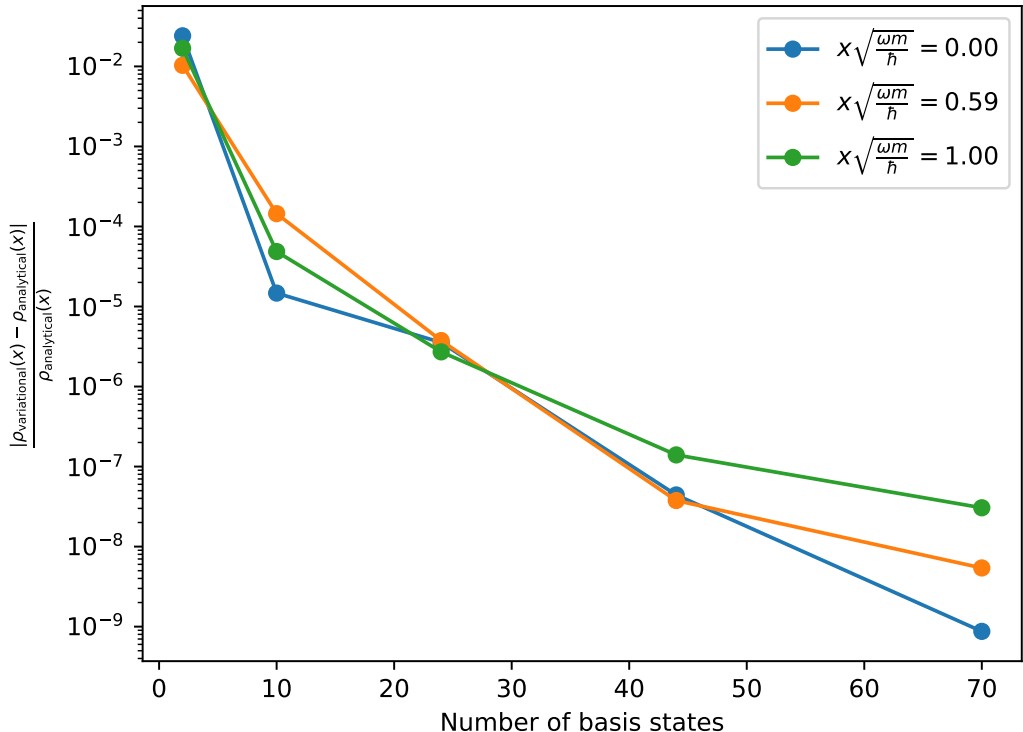

Figure 5: Detailed comparison between position space density in the 1+1 system at $g\sqrt{\frac{m}{\hbar^3\omega}} = 1$ at particular values of $x$ compared to the analytical result.

ing. Moreover, most numerical methods would perform worse the stronger the interaction strength is, but our method is exact at infinite interaction and thus works well both for small and strong interactions, with a peak of slower convergence at some intermediate interaction strength. Since our states are chosen such as to well approximate a state at finite interaction, the basis size is also relatively small and the computational power needed for the diagonalization is negligible. In particular, the method does not require sophisticated diagonalization algorithms or high performance computing tools, which is often the case for exact diagonalization methods. Note, moreover, that the matrix we diagonalize is not a particularly sparse matrix. Computing densities is more computationally heavy since they need to be evaluated for each position space coordinate separately, and thus scales linearly with the number of grid points. For majority densities, a double numerical integral must be performed which makes them heavier than the minority density. Furthermore, each position space coordinate for the minority density can be computed in parallel which can further decrease the computation time. While it is possible that our numerical techniques can be significantly improved, such as finding a way to efficiently parallelize the majority density calculations or optimize the grids on which we evaluate the numerical integrals, we will leave such investigations for future work. We also want to again stress that the time consuming calculations only have to be performed once for each chosen basis and we can then easily obtain the densities for any interaction strength $g$.

In this paper we only considered repulsive interactions. The method will not generalize trivially to attractive interaction since on the attractive sides, there are bound states that do not converge to any state when the interaction goes to infinite interaction strength, but instead

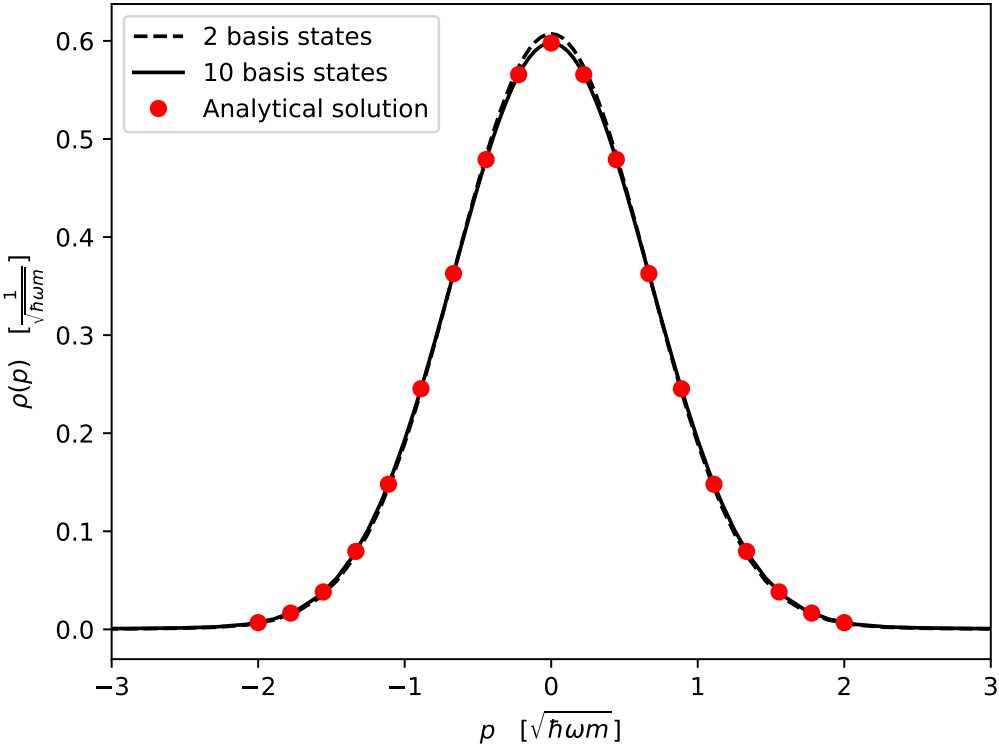

Figure 6: Momentum distribution of one of the particles in the ground state at $g\sqrt{\frac{m}{\hbar^3\omega}} = 1$ for the 1+1 system in the harmonic trap, compared with the analytical solution.

diverges with infinitely negative energy. Such states would not be well approximated by a linear combination of exact solutions at infinite interaction and states at zero interaction. It should be possible to extend the basis considered in this paper to states that would also capture these bound states, similarly to what was done in [51], but we will leave the study of attractive interactions for future work.

# 7   Acknowledgements

We would like to thank Artem Volosniev and Molte Andersen for useful discussions. REB acknowledges funding from Conselho Nacional de Desenvolvimento Científico e Tecnológico (CNPq) and Coordenação de Aperfeiçoamento de Pessoal de Nível Superior (CAPES). NTZ would like to acknowledge funding from Aarhus University Research Foundation under the JCS Fellowship program.

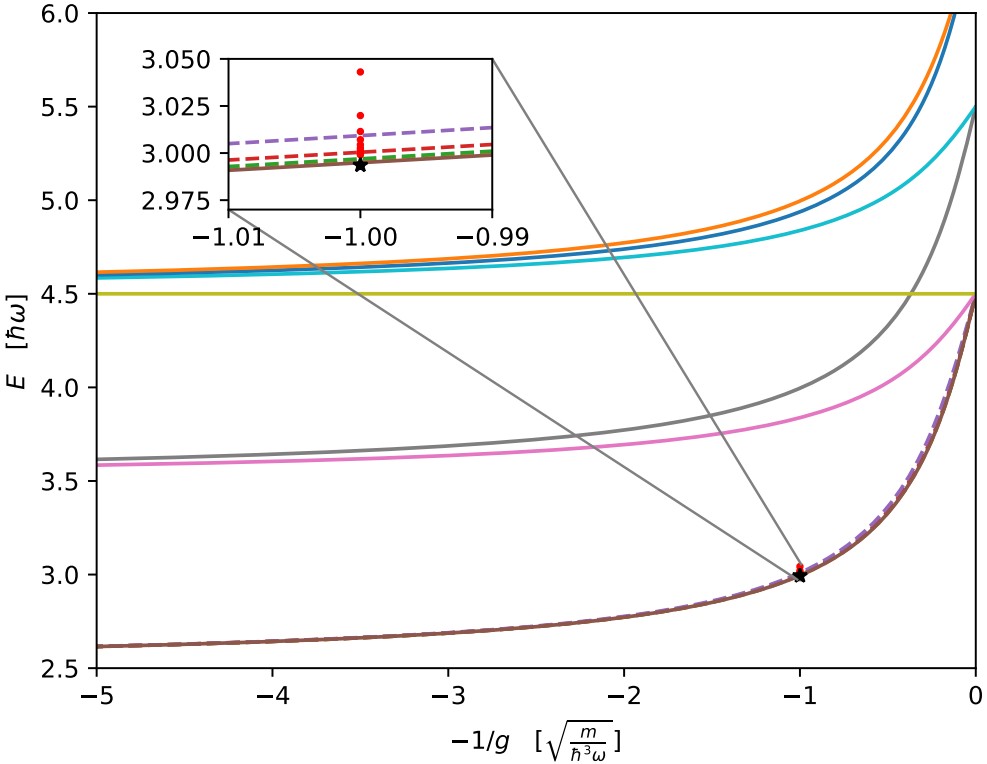

**Figure 7:** Energies for the lowest seven states for the 2+1 system computed using the variational method. The result using the matrix product states method for increasing accuracy is shown in red dots, with the black star being the extrapolated value. The energies are computed using basis states with basis threshold 8. The ground state is computed also using basis threshold 0, 2, 4 and 6 to show the convergence, which are shown with dashed lines.

## A  Some integral formulas

### A.1  Integral 1

In this section we will derive an expression for

$$I_{\vec{q},\vec{p}}^{k}(x) \equiv \int_{\mathcal{M}_k(x)} \Phi_{\vec{q}}(x_1,\ldots,x_n)\Phi_{\vec{p}}(x_1,\ldots,x_n), \tag{45}$$

where $\mathcal{M}_k(x)$ is the set where $x$ is smaller than exactly $k$ of the coordinates $x_1,\ldots,x_n$ and $\Phi_{\vec{v}}$ is the (normalized) totally antisymmetric wave function of the states corresponding to the quantum numbers in $\vec{v} = (v_1,\ldots,v_n)$. We will use induction to show that

$$I_{\vec{q},\vec{p}}^{k}(x) = \frac{1}{k!}\partial_\epsilon^k \det(A + \epsilon B)_{\epsilon=0}, \tag{46}$$

where $A$ is the matrix defined by $A_{ab}(x) = \int_{-\infty}^{x} f_{q_a}(x')f_{p_b}(x')\mathrm{d}x'$ and $B_{ab} = \int_{x}^{\infty} f_{q_a}(x')f_{p_b}(x')\mathrm{d}x' = \delta_{ab} - A_{ab}$. For $k=0$ we easily obtain

$$I_{\vec{q},\vec{p}}^{0}(x) = \int_{-\infty}^{x} \mathrm{d}x_1 \cdots \int_{-\infty}^{x} \mathrm{d}x_n \Phi_{\vec{q}}\Phi_{\vec{p}} = \det A(x), \tag{47}$$

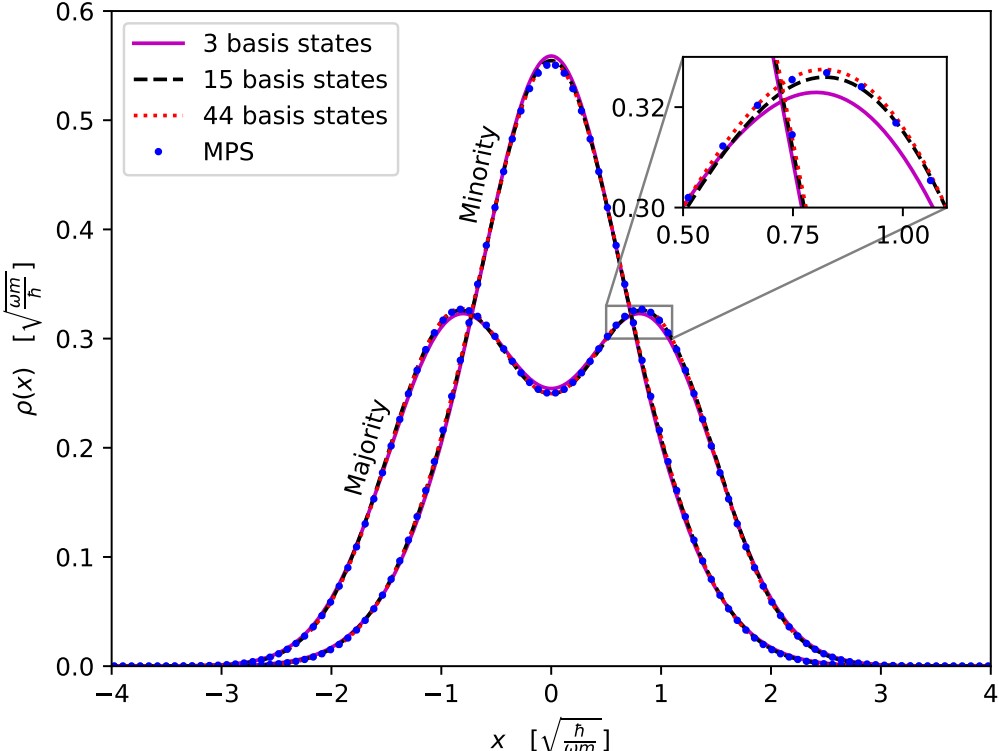

Figure 8: Position space density for the ground state at $g\sqrt{\frac{m}{\hbar^3\omega}} = 1$ for the 2+1 system in the harmonic potential, for different basis sizes compared to the matrix product states method. The density profile localized in the center is the minority density and the other one is the majority density.

which proves the base case. Now assume that $I^j_{\vec{q},\vec{p}}(x) = \frac{1}{j!}\partial^j_\epsilon \det(A+\epsilon B)_{\epsilon=0}$ for $j < k$. We then have

$$
\begin{aligned}
I^k_{\vec{q},\vec{p}}(x) &= \frac{1}{k}\sum_{i,j}(-1)^{i+j}\int_x^\infty f_{q_i}(x')f_{p_j}(x')\mathrm{d}x' I^{k-1}_{\vec{q}(i),\vec{p}(j)}(x)\\
&= \frac{1}{k}\sum_{i,j}(-1)^{i+j}\int_x^\infty f_{q_i}(x')f_{p_j}(x')\mathrm{d}x'\frac{1}{(k-1)!}\partial^{k-1}_\epsilon \det(A(i)(j)+\epsilon B(i)(j))_{\epsilon=0}\\
&= \frac{1}{k}\partial^{k-1}_\epsilon\left[\frac{1}{(k-1)!}\partial_\beta \det(A+\epsilon B+\beta B)_{\beta=0}\right]_{\epsilon=0}\\
&= \frac{1}{k!}\partial^k_\epsilon \det(A+\epsilon B)_{\epsilon=0},
\end{aligned}
\tag{48}
$$

where we have use the notation that $N(i)(j)$ is the matrix $N$ with row $i$ and column $j$ removed and similarly $\vec{q}(i)$ is the ordered set with the element indexed $i$ removed. We also used the formula $\mathrm{tr}[M\,\mathrm{adj}N] = \sum(-1)^{i+j}M_{ij}\det N(i)(j) = \partial_\epsilon\det(M+\epsilon N)_{\epsilon=0}$ and he factor $1/k = \binom{N}{k}/\left(N\binom{N-1}{k-1}\right)$ can be inferred from combinatorics and the normalization of the wavefunctions. Thus our formula is proven by induction.

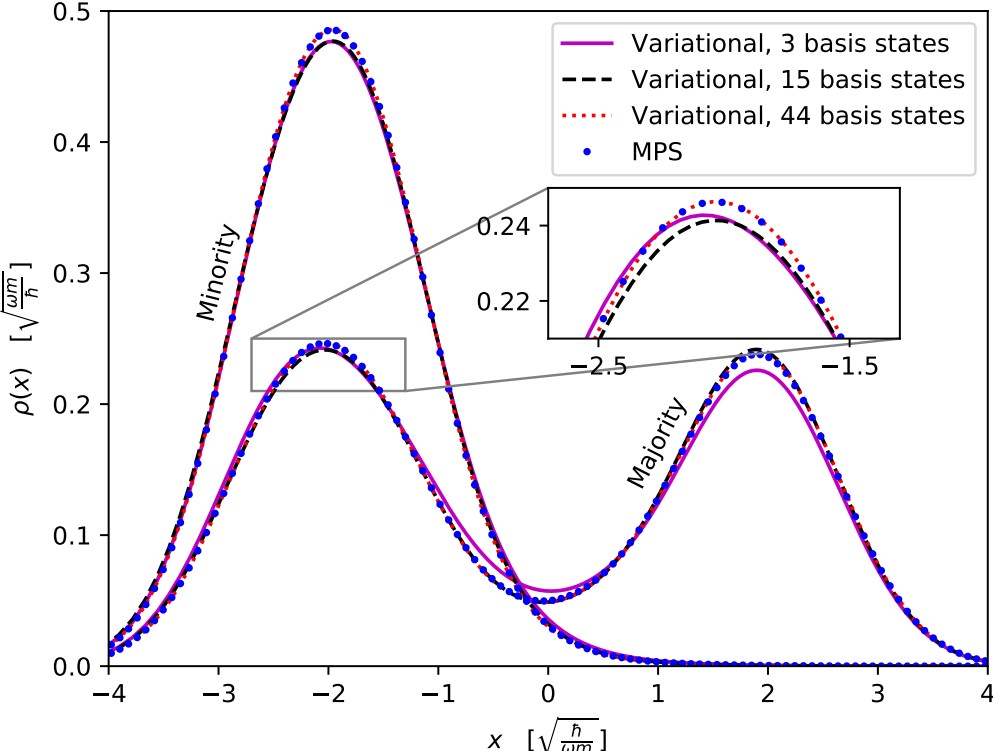

Figure 9: Position space density for the ground state at $g\sqrt{\frac{m}{\hbar^3\omega}} = 1$ for the 2+1 system in the double well potential in Figure 1, for different basis sizes compared with the matrix product states method. The profile localized in the left well is the minority density and the other one is the majority density.

## A.2   Integral 2

Let us now consider the integral

$$I_{\vec{q},\vec{p}}^{k,l}(x,x') \equiv \int_{\mathcal{M}_{k,l}(x,x')} \Phi_{\vec{q}}(x_1,\dots,x_n)\Phi_{\vec{p}}(x_1,\dots,x_n), \tag{49}$$

where $\mathcal{M}_{k,l}(x,x')$ is the set where $x$ is smaller than exactly $k$ of the coordinates $x_1,\dots,x_n$ and $x'$ is smaller than exactly $l$ of the coordinates $x_1,\dots,x_n$. $\Phi_{\vec{v}}$ is the (normalized) totally antisymmetric wave function of the states corresponding to the quantum numbers in $\vec{v} = (v_1,\dots,v_n)$. The result is

$$I_{n,m}^{k,l}(x,x') = \frac{1}{|k-l|!(min(k,l))!}\partial_{\epsilon}^{|k-l|}\partial_{\nu}^{min(k,l)}\det(A + \nu B + \epsilon C)_{\epsilon=0,\nu=0}, \tag{50}$$

where $A_{ij} = \int_{-\infty}^{min(x,x')} f_{n_i}(x'')f_{m_j}(x'')dx''$, $C_{ij} = \int_{min(x,x')}^{max(x,x')} f_{n_i}(x'')f_{m_j}(x'')dx''$ and $B_{ij} = \int_{max(x,x')}^{\infty} f_{n_i}(x'')f_{m_j}(x'')dx''$. We can also prove this by induction. Note that if we assume $x > x'$ and $k = 0$, the formula is the same as (46) if the upper integral limit is changed from $\infty$ to $x$ and the same proof goes through. We will thus use this as a base case for our induction proof and thus assuming without loss of generality that $x > x'$, we can prove the formula for $l,k$ with $k < l$ by assuming that it holds for $k-1,l-1$. Following the exact same

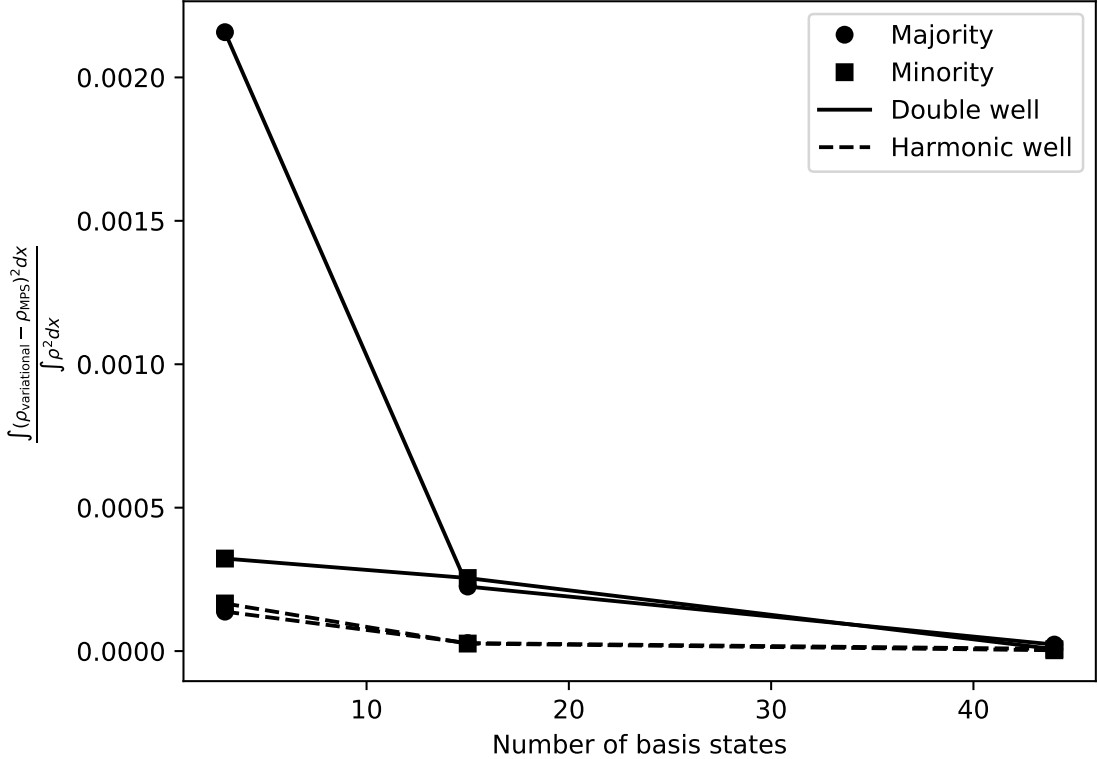

Figure 10: Integrated difference square of the position space density for the ground state at $g\sqrt{\frac{m}{\hbar^3\omega}} = 1$ for the 2+1 system compared with the matrix product states method.

reasoning as in the proof in A.1, we have

$$
\begin{aligned}
I_{\vec{q},\vec{p}}^{k,l}(x,x') &= \frac{1}{k}\sum_{i,j}(-1)^{i+j}\int_x^\infty f_{q_i}(x'')f_{p_j}(x'')\mathrm{d}x'' I_{\vec{q}(i),\vec{p}(j)}^{k-1,l-1}(x,x') \\
&= \frac{1}{k}\sum_{i,j}(-1)^{i+j}\int_x^\infty f_{q_i}(x'')f_{p_j}(x'')\mathrm{d}x'' \\
&\quad \frac{1}{|k-l|!k!}\partial_\epsilon^{|k-l|}\partial_\nu^{k-1}\det(A(i)(j)+\nu B(i)(j)+\epsilon C(i)(j))_{\epsilon=0,\nu=0} \\
&= \frac{1}{k}\partial_\epsilon^{|k-l|}\partial_\nu^{k-1}\left[\frac{1}{|k-l|!(k-1)!}\partial_\beta\det(A+\nu B+\epsilon C+\beta B)_{\beta=0}\right]_{\epsilon=0,\nu=0} \\
&= \frac{1}{|l-k|!k!}\partial_\epsilon^{|k-l|}\partial_\nu^k\det(A+\nu B+\epsilon C)_{\epsilon=0,\nu=0}.
\end{aligned}
\tag{51}
$$

Since we assumed that $k<l$ and $x>x'$, and the exact same proof can be done for $k>l$ and $x<x'$, formula (50) follows.

## B  Two particle system

In this section we review the analytical solution of two particles in a harmonic trap, with a delta function interaction [54]. The full Hamiltonian is

$$
H = \frac{1}{2}x_1^2 + \frac{1}{2}x_2^2 + \frac{1}{2}p_1^2 + \frac{1}{2}p_2^2 + V,
\tag{52}
$$

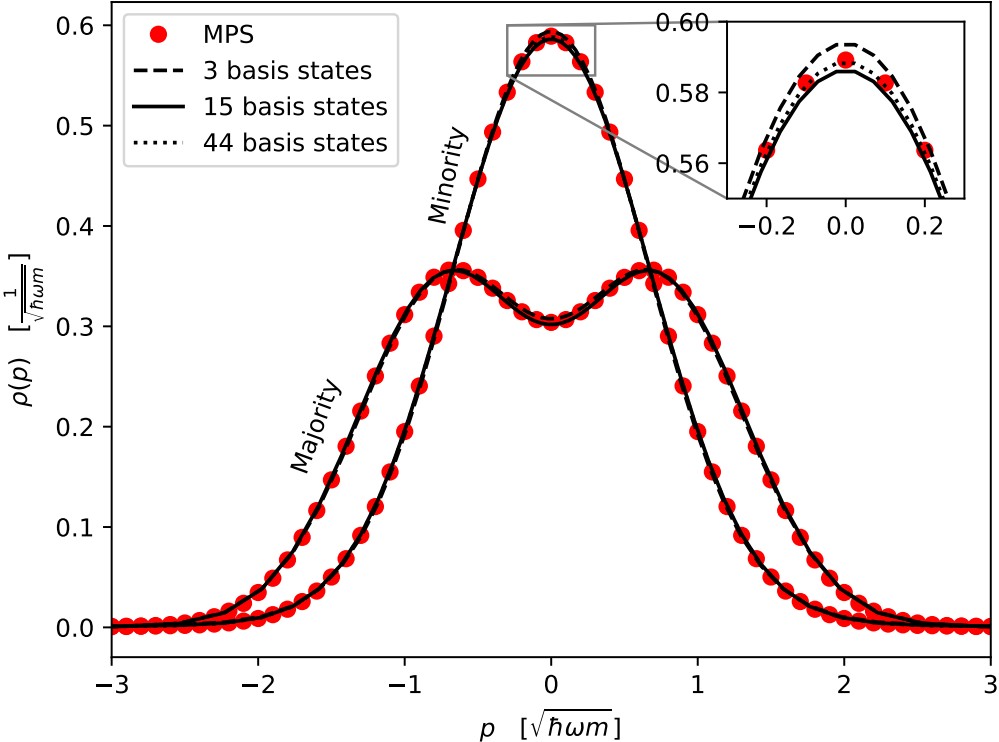

Figure 11: Momentum space density of the ground state at $g\sqrt{\frac{m}{\hbar^3\omega}} = 1$ for the 2+1 system in the harmonic trap, compared with the matrix product states method

where

$$\langle x_1, x_2|V|x_1', x_2'\rangle = g\delta(x_1 - x_2)\delta(x_1 - x_1')\delta(x_2 - x_2'). \tag{53}$$

By introducing Jacobi coordinates $x = (x_1 - x_2)/\sqrt{2}$, $p = (p_1 - p_2)/\sqrt{2}$, $X = (x_1 + x_2)/\sqrt{2}$ and $P = (p_1 + p_2)/\sqrt{2}$ we can split this Hamiltonian into two parts, namely

$$H = H_{\text{rel}} + H_{\text{CM}}, \tag{54}$$

where $H_{\text{CM}} = X^2/2 + P^2/2$ is just a harmonic oscillator corresponding to the center-of-mass motion, and

$$H_{\text{rel}} = \frac{x^2}{2} + \frac{p^2}{2} + \frac{g}{\sqrt{2}}\delta(x)\delta(x - x'). \tag{55}$$

The hard part, which will occupy most of this appendix, is solving for the eigenstates of $H_{\text{rel}}$. The full set of eigenstates and eigenenergies are then obtained by tensor product with the eigenstates of $H_{\text{CM}}$.

We will solve for the wavefuctions by first expanding in a harmonic oscillator basis. The Harmonic oscillator eigenfunctions are given by

$$f_n(x) = \frac{1}{\sqrt{2^n n!}}\pi^{-1/4}e^{-\frac{x^2}{2}}H_n(x), \tag{56}$$

where $H_n$ are the Hermite polynomials. The energy is given by $E_n = n + 1/2$. Let $|\Phi\rangle$ be an

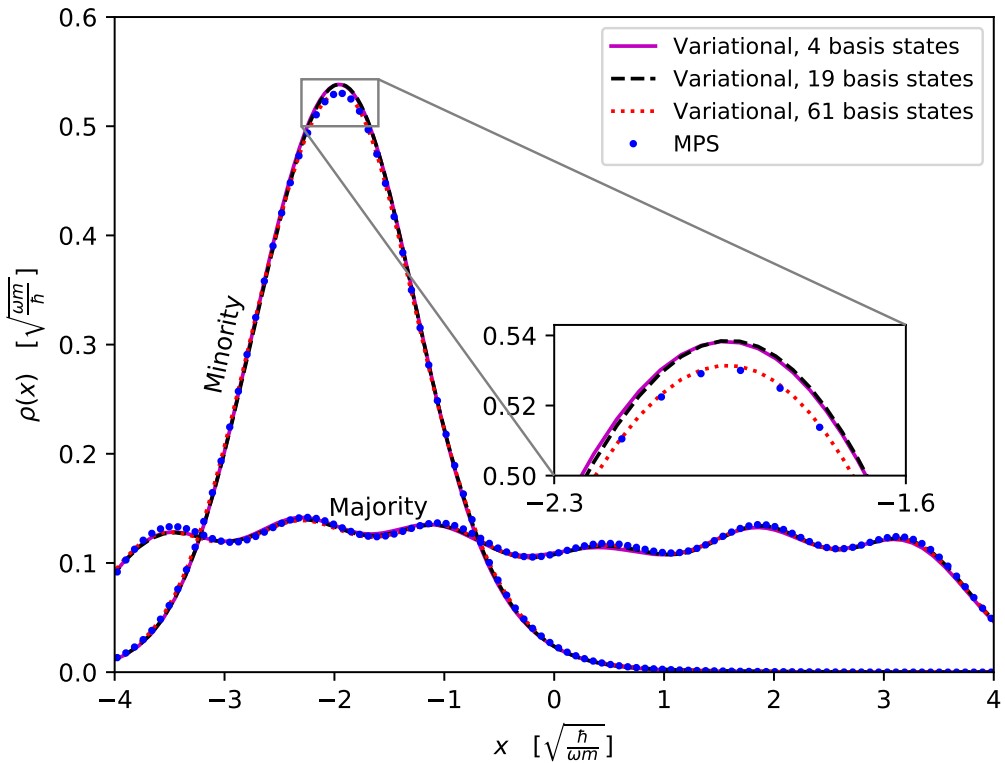

Figure 12: Position space density for the ground state at $g\sqrt{\frac{m}{\hbar^3\omega}} = 1$ in the double well potential in Figure 1 for the 6+1 system. The density profile localized in the left well is the minority density and the other one is the majority density.

eigenstate for $H_{\text{rel}}$. We have

$$H_{\text{rel}}|\Phi\rangle = E_\Phi|\Phi\rangle \Rightarrow E_n\langle n|\Phi\rangle + \sum_{m=0}^{\infty}\langle n|V|m\rangle\langle m|\Phi\rangle = E_\Phi|\Phi\rangle. \tag{57}$$

Solving for $c_n \equiv \langle n|\Phi\rangle$ and defining the quantity $A = \sum f_n(0)c_n$ we obtain

$$c_n = \frac{g}{\sqrt{2}}f_n(0)\frac{A}{E_\Phi - E_n}. \tag{58}$$

Now multiplying both sides by $f_n(0)$ and summing over $n$, we can cancel $A$ from both sides to obtain

$$1 = \frac{g}{\sqrt{2}}\sum_n\frac{f_n(0)^2}{E_\Phi - 1/2 - n} = \frac{g}{\sqrt{2}}\sum_n\frac{f_{2n}(0)^2}{E_\Phi - 1/2 - 2n}. \tag{59}$$

For the case where $A = 0$, for which we can not cancel it from both sides to obtain equation (59), see Appendix B.1. For the Hermite polynomials, we have $H_n(0) = 0$ if $n$ is odd, and $H_{2n}(0) = (-1)^n(2n!)/n!$, which is the reason why we have omitted the odd terms. The wavefunction is given by a similar formula, namely

$$\Phi(x) = \frac{g}{\sqrt{2}}A\sum_n\frac{f_{2n}(0)f_{2n}(x)}{E_\Phi - 2n - 1/2}. \tag{60}$$

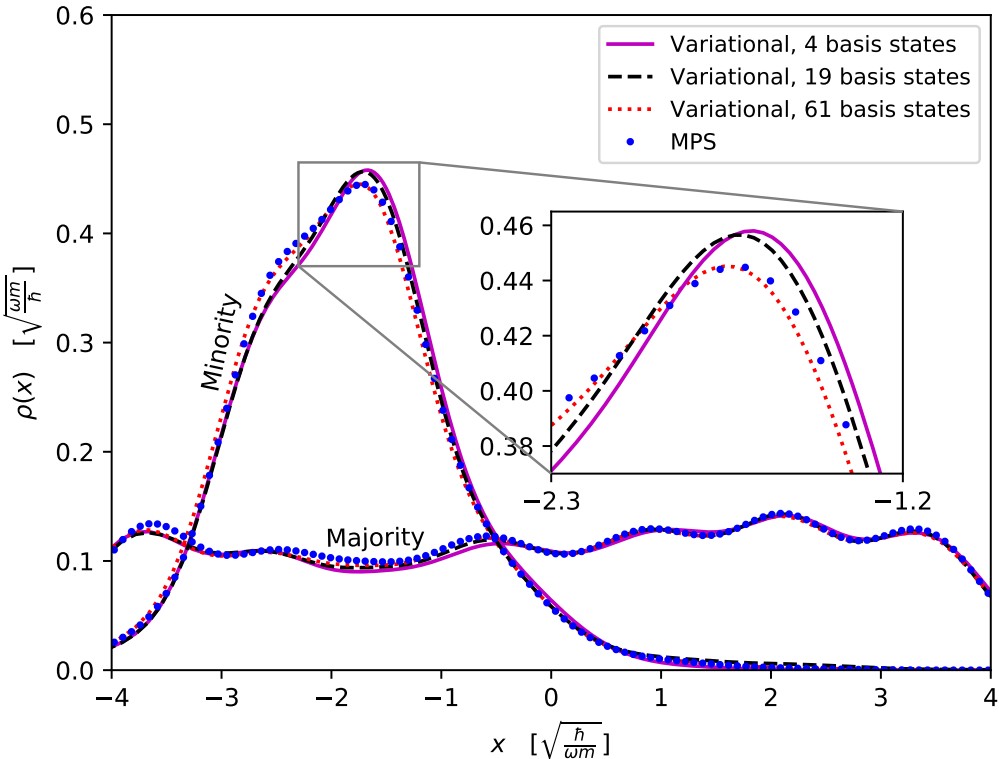

Figure 13: Position space density for the ground state at $g\sqrt{\frac{m}{\hbar^3\omega}} = 10$ in the double well potential in Figure 1 for the 6+1 system. The density profile localized in the left well is the minority density and the other one is the majority density.

It thus makes sense to treat these simultaneously, so let us define

$$\mathcal{F}(x) = \sqrt{\pi} \sum_n \frac{f_{2n}(0) f_{2n}(x)}{n - \nu}. \tag{61}$$

To compute this function, we use the following relation between Hermite polynomials and Laguerre polynomials

$$H_{2n}(x) = (-1)^n 2^{2n} n! L_n^{-1/2}(x^2). \tag{62}$$

We thus obtain

$$\mathcal{F}(x) = \sum_n \frac{e^{-\frac{x^2}{2}} L_n^{-1/2}(x^2)}{n - \nu}. \tag{63}$$

Now we use the integral representation

$$\frac{1}{n - \nu} = \int_0^\infty dy \frac{1}{(1+y)^2} \left(\frac{y}{1+y}\right)^{n-\nu-1}, \tag{64}$$

to obtain

$$\mathcal{F}(x) = \int_0^\infty \frac{dy}{(1+y)^2} \left(\frac{y}{1+y}\right)^{-\nu-1} e^{-x^2/2} \sum_n L_n^{-1/2}(x^2) \left(\frac{y}{1+y}\right)^n. \tag{65}$$

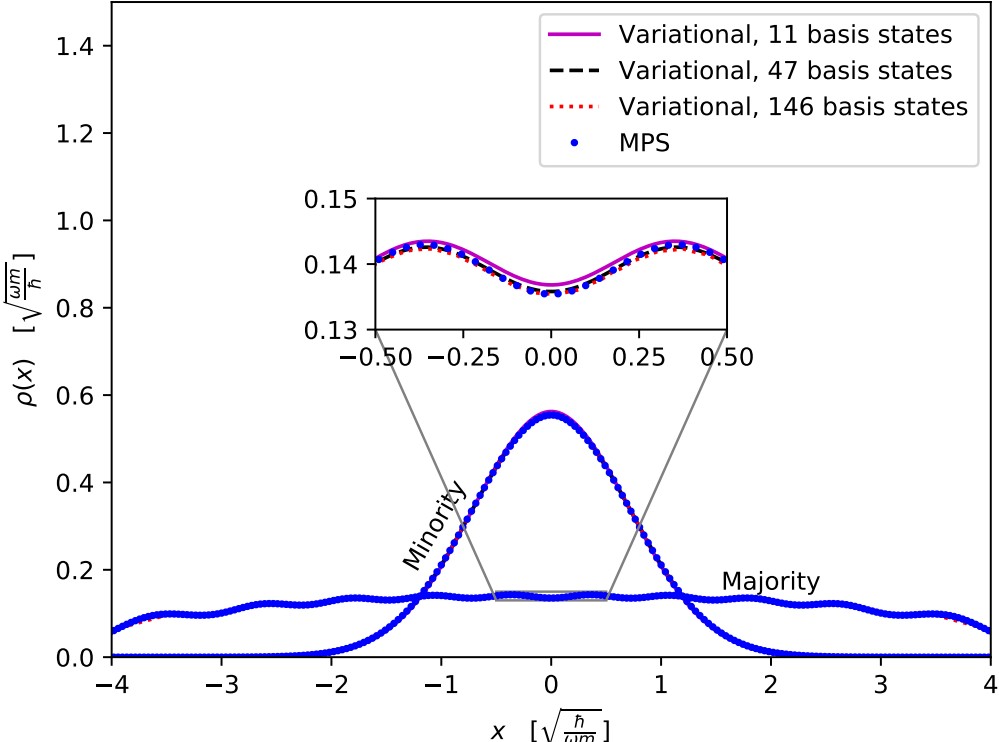

Figure 14: Position space density for the ground state at $g\sqrt{\frac{m}{\hbar^3\omega}} = 1$ in the harmonic potential.

Now we can recognize the generating function $e^{-tx/(1-t)}(1-t)^{-\alpha-1} = \sum t^n L_n^\alpha(x)$ to obtain

$$\mathcal{F}(x) = e^{-x^2/2}\int_0^\infty dy(1+y)^{\nu-1/2}y^{-\nu-1}e^{-yx^2} = \Gamma(-\nu)e^{-x^2/2}U(-\nu,1/2,x^2), \qquad (66)$$

where we have used a standard representation for the confluent hypergeometric function $U$. At $x = 0$, we can use the relation $U(-\nu,1/2,0) = \Gamma(1/2)/\Gamma(1/2-\nu) = \sqrt{\pi}/\Gamma(1/2-\nu)$, to obtain

$$\mathcal{F}(0) = \sqrt{\pi}\frac{\Gamma(-\nu)}{\Gamma(\frac{1}{2}-\nu)}. \qquad (67)$$

Thus for the energy, we must solve the equation

$$1 = -g\frac{\mathcal{F}_{\nu=E_\Phi/2-1/4}(0)}{2\sqrt{2\pi}} = -\frac{g}{2\sqrt{2}}\frac{\Gamma(-E_\Phi/2+1/4)}{\Gamma(-E_\Phi/2+3/4)}. \qquad (68)$$

For the wavefunction, we instead have

$$\Phi(x) = -\frac{gA}{2\sqrt{2\pi}}\mathcal{F}_{\nu=E_\Phi/2-1/4}(x) = -\frac{gA}{2\sqrt{2\pi}}\Gamma(-E_\Phi/2+1/4)e^{-x^2/2}U(-E_\Phi/2+1/4,1/2,x^2). \qquad (69)$$

To find the normalization constant $A$ we can consider the normalization constraint

$$1 = \sum_n c_n^2 = \frac{g^2}{2} A^2 \sum_n \frac{f_n^2(0)}{(E_\Phi - n - 1/2)^2}$$
$$= \frac{g^2}{4} A^2 \partial_{E_\Phi} \frac{\Gamma(-E_\Phi/2 + 1/4)}{\Gamma(-E_\Phi/2 + 3/4)}. \tag{70}$$

Defining $\psi(x) = \Gamma'(x)/\Gamma(x)$ and using the energy formula (68), we can simplify this to

$$A^2 = \frac{2\sqrt{2}}{g(\psi(-E_\Phi/2 + 1/4) - \psi(-E_\Phi/2 + 3/4))}. \tag{71}$$

### B.1  Odd states

What we have obtained so far are all even parity states where the wavefunction in position space is an even function. The odd parity states are just odd harmonic oscillator states and they are unaffected by the interaction since they vanish at $x = 0$. These states would have $A = 0$ and thus the step to obtain equation (59) would be illegitimate.

### B.2  Absolute coordinates

The full eigenstates are then obtained by also multiplying by the center of mass states. The complete wave function for $H = H_{\text{rel}} + H_{\text{CM}}$ is given by

$$\Phi_{k,n}(x, X) = \Phi_k(x) f_n(X), \tag{72}$$

where we have labeled all eigenstates of $H_{\text{rel}}$ (both even and odd) by $\Phi_k$ for $k = 0, 1, \dots$ and $f_n$ are just the standard harmonic oscillator wavefunctions. The energy is likewise $E_{k,n} = E_{\Phi_k} + E_n$ where $E_n = n + 1/2$ is the nth harmonic oscillator energy.

To compare with the variational method in this paper, we would also like to compute the coordinate and momentum densities. Recall that $x = (x_1 - x_2)/\sqrt{2}$ and $X = (x_1 + x_2)/\sqrt{2}$. The single particle density matrix is just the square of the wavefunction in absolute coordinates, namely

$$\rho(x_1, x_2) = \Phi_{k,n}^2(\frac{x_1 - x_2}{\sqrt{2}}, \frac{x_1 + x_2}{\sqrt{2}}), \tag{73}$$

and the density is thus given by

$$\rho(x_1) = \int_{-\infty}^{\infty} \Phi_{k,n}^2(\frac{x_1 - x_2}{\sqrt{2}}, \frac{x_1 + x_2}{\sqrt{2}}) dx_2. \tag{74}$$

The momentum density can then be obtained by

$$\rho(p) = \frac{1}{2\pi} \int_{-\infty}^{\infty} \int_{-\infty}^{\infty} dx_1 dx_2 e^{ip(x_1 - x_2)} \rho(x_1, x_2). \tag{75}$$

## C  Wavefunctions and energies for a smooth double well potential

In this appendix we give details on energies and wavefunctions of the double well potential. The double well potential is defined as

$$V(x) = \begin{cases} \frac{1}{2}\omega_0^2(x - x_0)^2 + \Delta_0 & x < x_L < 0, \\ -\frac{1}{2}\omega_1^2(x - x_1)^2 + \Delta_1 & x_1 < x < x_2, \\ \frac{1}{2}\omega_2^2(x - x_2)^2 + \Delta_2 & x > x_2 > 0, \end{cases} \tag{76}$$

where we require $x_0 < x_1 < x_2$ and $x_L < x_R$. Continuity of the potential as well as its derivatives at two points $x_L$ and $x_R$ implies the equations

$$\frac{1}{2}(x_L - x_0)^2 \omega_0^2 + \Delta_0 = -\frac{1}{2}(x_L - x_1)^2 \omega_1^2 + \Delta_1, \tag{77}$$

$$-\frac{1}{2}(x_R - x_1)^2 \omega_1^2 + \Delta_1 = \frac{1}{2}(x_R - x_2)^2 \omega_2^2 + \Delta_2, \tag{78}$$

$$\omega_0^2(x_L - x_0) = -\omega_1^2(x_L - x_1), \tag{79}$$

$$-\omega_1^2(x_R - x_1) = \omega_2^2(x_R - x_2). \tag{80}$$

This system is uniquely solved for $\omega_1$, $x_1$, $x_L$ and $x_R$ given the physically relevant quantities $\omega_0$, $\omega_2$, $x_0$, $x_2$, $\Delta_0$, $\Delta_1$ and $\Delta_2$. The solution is given by

$$\omega_1^{-2} = \frac{1}{2(\Delta_2 - \Delta_0)^2 \omega_0^4 \omega_2^4}$$
$$\Big[ -2\sqrt{(x_2 - x_0)^2 (\Delta_1 - \Delta_0)(\Delta_1 - \Delta_2) \omega_0^6 \omega_2^6 ((x_2 - x_0)^2 \omega_0^2 \omega_2^2 + 2(\Delta_2 - \Delta_0)(\omega_0^2 - \omega_2^2))}$$
$$- 2(\Delta_1 - \Delta_0)(\Delta_2 - \Delta_0)\omega_0^2 \omega_2^4 - \omega_0^4 \omega_2^2 (2(\Delta_2 - \Delta_0)(\Delta_2 - \Delta_1)$$
$$+ (x_2 - x_0)^2(-2(\Delta_1 - \Delta_0) + (\Delta_2 - \Delta_0))\omega_2^2)\Big], \tag{81}$$

$$x_1 - x_0 = \frac{1}{(\Delta_2 - \Delta_0)}\Big[ (x_2 - x_0)(\Delta_1 - \Delta_0) - \frac{1}{(x_2 - x_0)\omega_0^4 \omega_2^4} \times$$
$$\sqrt{(x_2 - x_0)^2 (\Delta_1 - \Delta_0)(\Delta_1 - \Delta_2)\omega_0^6 \omega_2^6 ((x_2 - x_0)^2 \omega_0^2 \omega_2^2 + 2(\Delta_2 - \Delta_0)(\omega_0^2 - \omega_2^2))}\Big] \tag{82}$$

and then $x_L$ and $x_R$ are given by

$$x_L = \frac{x_0 \omega_0^2 + x_1 \omega_1^2}{\omega_0^2 + \omega_1^2} \tag{83}$$

$$x_R = \frac{x_2 \omega_2^2 + x_1 \omega_1^2}{\omega_2^2 + \omega_1^2}. \tag{84}$$

Extra care for these formulas must be taken when evaluating these expressions for $\Delta_0 = \Delta_2$. In this case we have

$$\omega_1^2 = \frac{8(\Delta_1 - \Delta_0)(x_0 - x_2)^2 \omega_0^4 \omega_2^4}{(x_2 - x_0)^4 \omega_0^4 \omega_2^4 + 4(\Delta_1 - \Delta_0)^2(\omega_2^2 - \omega_0^2)^2 - 4(\Delta_1 - \Delta_0)(x_2 - x_0)^2 \omega_0^2 \omega_2^2(\omega_0^2 + \omega_2^2)} \tag{85}$$

$$x_1 = \frac{2(\Delta_1 - \Delta_0)\omega_0^2 + (2\Delta_0 - 2\Delta_1 + (x_0^2 - x_2^2)\omega_0^2)\omega_2^2}{2(x_0 - x_2)\omega_0^2 \omega_2^2}. \tag{86}$$

For the symmetric case (symmetric around $x_1 = (x_0 + x_2)/2$) where we also have $\omega_0 = \omega_2$, we have

$$\omega_1^2 = \frac{8(\Delta_1 - \Delta_0)\omega_0^2}{8(\Delta_0 - \Delta_1) + (x_0 - x_2)^2 \omega_0^2}. \tag{87}$$

We can compute an upper limit on the parameter $\Delta_1$. The highest value is the value such that $\omega_1 = \infty$, namely we have the more well known double well potential which has a discontinuous derivative between the wells. For such a potential the discontinuity is at the intersection of the left and right wells, namely we solve

$$\frac{1}{2}(x_M - x_0)^2 \omega_0^2 + \Delta_0 = \frac{1}{2}(x_M - x_1)^2 \omega_2^2 + \Delta_2, \tag{88}$$

which results in the solution

$$x_M = \frac{x_0 \omega_0^2 - x_2 \omega_2^2 + \sqrt{2(\Delta_2 - \Delta_0)\omega_0^2 + (2\Delta_0 - 2\Delta_2 + (x_0 - x_2)^2 \omega_0^2)\omega_2^2}}{\omega_0^2 - \omega_2^2}. \tag{89}$$

Then the upper limit of $\Delta_1$ is given by $\Delta_{1,\max} = \frac{1}{2}(x_M - x_0)^2 \omega_0^2 + \Delta_0$.

We will now work out the wavefunctions and energies. We will work in units where $\hbar = 1$ and $m = 1$ (the mass of the particle) and for simplicity and we will define $\nu_0$ and $\nu_2$ by $E = \omega_0(\nu_0 + \frac{1}{2}) + \Delta_0 = \omega_2(\nu_2 + \frac{1}{2}) + \Delta_2$. The eigenfunctions are now uniquely given by

$$\psi(x) = C_0 D_{\nu_0}\left(-\sqrt{2\omega_0}(x - z_0)\right) \tag{90}$$

for $x < x_L$ and

$$\psi(x) = C_2 D_{\nu_2}\left(\sqrt{2\omega_2}(x - z_2)\right) \tag{91}$$

for $x > x_R$ and for some constants $C_0, C_2$ (this follows since these are the only solutions with the correct falloffs at $x \to \pm\infty$). The function $D$ is the parabolic cylinder function given by

$$D_\nu(z) = 2^{\nu/2} e^{-z^2/4} \left[ \frac{\Gamma(\frac{1}{2})}{\Gamma(\frac{1-\nu}{2})} {}_1F_1\left(-\frac{\nu}{2}; \frac{1}{2}; \frac{z^2}{2}\right) + \frac{z}{\sqrt{2}} \frac{\Gamma(-\frac{1}{2})}{\Gamma(-\frac{\nu}{2})} {}_1F_1\left(\frac{1-\nu}{2}; \frac{3}{2}; \frac{z^2}{2}\right) \right], \tag{92}$$

where ${}_1F_1$ is the confluent hypergeometric function. Note that this function is a linear combination of the two linearly independent solution of the Schrödinger equation in a harmonic well, and the relative coefficient has been fixed by requiring falloff at infinity. In the intermediate region we need to solve the Schrödinger equation for an inverted harmonic well. It can be showed that the solution then is

$$\psi(x) = C_1^{(1)} K_{\nu_1}^{(1)}(\sqrt{2\omega_1}(x - x_1)) + C_1^{(2)} K_{\nu_1}^{(2)}(\sqrt{2\omega_1}(x - x_1)), \tag{93}$$

where

$$K_\nu^{(1)}(z) = e^{-iz^2/4} {}_1F_1(\frac{i\nu}{2} + \frac{i}{4} + \frac{1}{4}; \frac{1}{2}; \frac{iz^2}{2}) \tag{94}$$

and

$$K_\nu^{(2)}(z) = e^{-iz^2/4} z\, {}_1F_1(\frac{i\nu}{2} + \frac{i}{4} + \frac{3}{4}; \frac{3}{2}; \frac{iz^2}{2}) \tag{95}$$

and where we have parametrized the energy as $E = \omega_1(\nu_1 + \frac{1}{2}) + \Delta_1$ (which we recall is also equal to $\omega_0(\nu_0 + \frac{1}{2}) + \Delta_0 = \omega_2(\nu_2 + \frac{1}{2}) + \Delta_2$). Despite the complex arguments, these are real functions. These solutions should now be glued smoothly across the points $x_L$ and $x_R$ such that $\psi$ and $\psi'$ are continuous. To simplify the equations, we will define $r = \omega_2/\omega_1$, $R = \omega/\omega_1$,

$\Delta = \hbar\omega_1\delta$, $C = \hbar\omega_1 c$ and we work in units where $\mu\omega_1/\hbar = 1$. This gives the equations

$$C_0 D_{\nu_0}\left(-\sqrt{2\omega_0}(x_L - x_0)\right) = C_1^{(1)} K_{\nu_1}^{(1)}(\sqrt{2\omega_1}(x_L - x_1)) + C_1^{(2)} K_{\nu_1}^{(2)}(\sqrt{2\omega_1}(x_L - x_1)),$$
(96)

$$C_2 D_{\nu_2}\left(\sqrt{2\omega_2}(x_R - x_2)\right) = C_1^{(1)} K_{\nu_1}^{(1)}(\sqrt{2\omega_1}(x_R - x_1)) + C_1^{(2)} K_{\nu_1}^{(2)}(\sqrt{2\omega_1}(x_R - x_1)),$$
(97)

$$-\sqrt{\omega_0}C_0 D'_{\nu_0}\left(-\sqrt{2}(x_L - x_0)\right) = \sqrt{\omega_1}C_1^{(1)} K_{\nu_1}^{(1)'}(\sqrt{2\omega_1}(x_L - x_1))$$
$$+ \sqrt{\omega_1}C_1^{(2)} K_{\nu_1}^{(2)'}(\sqrt{2\omega_1}(x_L - x_1)),$$
(98)

$$\sqrt{\omega_2}C_2 D'_{\nu_2}\left(-\sqrt{2}(x_R - x_2)\right) = \sqrt{\omega_1}C_1^{(1)} K_{\nu_1}^{(1)'}(\sqrt{2\omega_1}(x_R - x_1))$$
$$+ \sqrt{\omega_1}C_1^{(2)} K_{\nu_1}^{(2)'}(\sqrt{2\omega_1}(x_R - x_1)).$$
(99)

If we are given $\nu_0, \nu_1, \nu_2$ (which are all determined by the energy $E$), this is a linear system of equations for $C_0, C_2, C_1^{(1)}, C_1^{(2)}$. For this system to have a non-trivial solution, the determinant of the corresponding matrix must vanish and this condition is what determines the energy (or equivalently the parameters $\nu_0, \nu_1, \nu_2$). This system of equations, supplemented with normalization of the wave function, then fixes all the constants $C_0, C_2, C_1^{(1)}, C_1^{(2)}$. In general, if we piece together $N$ different quadratic (or other analytically solvable) potentials, the energy will be obtained by solving the equation resulting from enforcing zero determinant of a $2(N-1) \times 2(N-1)$ matrix.

# D  Matrix Product States

Throughout this work we compare our analytical method with simulations performed with Matrix Product States (MPS), using the Open Source MPS (OSMPS) libraries [52]. In these calculations, we employ the Hubbard model as an approximation to the continuum in order to obtain static properties of a fermionic polaron system. Thus the spinful lattice Hamiltonian is written as

$$H = -t\sum_{j,\sigma}(c_{j+1,\sigma}^\dagger c_{j,\sigma} + \text{H.c.}) + U\sum_j n_{j,\uparrow}n_{j,\downarrow} + \sum_{j,\sigma}\epsilon_j n_{j,\sigma},$$
(100)

where $c^\dagger$ and $c$ are the creation and annihilation operators, respectively, $t$ is the hopping parameter and $U$ denotes the strength of the on-site interactions between fermions with different spin projections. We denote the internal states as $|\uparrow\rangle$ for the background fermions and $|\downarrow\rangle$ for the impurity. Since we consider only a single $|\downarrow\rangle$ fermion, we have naturally $\sum_j n_{j,\downarrow} = 1$, with $\sum_j n_{j,\uparrow}$ also being normalized to the number of background fermions. We include additionally the trapping potential as the position-dependent $\epsilon_j$ parameter.

We simulate the continuum by taking a total of $L = 256$ sites. We thus obtain a lattice spacing $a = l/L$ where $l$ is the total length assumed for the trapping potential. The hopping parameter is related to the kinetic term in the continuum as $t = 1/(2ma^2)$, where $m$ is the atomic mass, which we take to be 1. The continuum and discrete interaction parameters are related as $U = g/a$. To obtain matching energies, we must include an additional term in the Hamiltonian given by $\sum_j 1/a^2$. In some cases, to improve the accuracy we compute the results for several increasing values of $L$ and then extrapolate to a final value using a function of the form $f(L) = A + B/L + C/L^2$.

# E   Polynomial interpolation for computing determinants

At several stages in the technique used in this paper we have to compute derivatives of determinants of the form $\partial_\epsilon^i D(\epsilon)|_{\epsilon=0} = \partial_\epsilon^i \frac{1}{i!} \det(M(\epsilon))|_{\epsilon=0}$, where $M(\epsilon)$ is some $n \times n$ matrix and $i = 0, \ldots, n$. We evaluate these derivatives by computing the function $D(\epsilon)$ on $n+1$ values with $\epsilon_i = -1 + 2i/n$, $i = 0, \ldots, n$, and then fitting a polynomial to these values and extracting the coefficients. These coefficients can be obtained by multiplying the vector $D(\epsilon_i)$ with the inverse of the matrix $K_{ij} \equiv \epsilon_i^j$.

For the single-particle density matrix, we also need to compute terms of the form $\partial_\epsilon^i \partial_\delta^j D(\epsilon, \delta)|_{\epsilon=0, \delta=0}$. This is done similarly be fitting a polynomial of two variables to the values $D(\epsilon_i, \epsilon_j)$ with $\epsilon_i = -1 + 2i/n$, $i = 0, \ldots, n$. We carry out the polynomial fit by applying the $(n+1)^2 \times (n+1)^2$ matrix $K_{IJ} \equiv \epsilon_{\lfloor J/(n+1) \rfloor}^{\lfloor J/(n+1) \rfloor} \epsilon_{I \bmod (n+1)}^{J \bmod (n+1)}$ on the $(n+1)^2$ vector $D_I = D(\epsilon_{\lfloor I/(n+1) \rfloor}, \epsilon_{I \bmod (n+1)})$.

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
