# Peer review of "A systematic interpolatory method for an impurity in a one-dimensional fermionic background"

_SciPost Physics, doi:SciPost Phys. 9, 005 (2020)_

## Round 1 · Referee Report · Anonymous (Referee 1) · 2019-12-6

Report

This paper describes a (numerical) variational procedure to approximately solve the $1+N_{\uparrow}$ fermionic problem in a harmonic trap in one dimension. The basic idea is to consider a number of eigenstates of the non-interacting system and a number of eigenstates in the strongly interacting limit, and use these to construct a new basis via the Gram-Schmidt procedure in which the Hamiltonian is then diagonalized. Although there are methods available which can (in principle) give the answer to this problem free from any systematic error, the advantage of this method is that it is less time-consuming (one needs to construct the basis once to get results for all values of the coupling strength). The results for the 1+1 and 1+2 problem convincingly show that the method can provide very accurate results. As explained below, my main concern is the fact that the number of particles remains small.

I believe the paper can be considered for publication in Scipost after some revisions and after some questions have been addressed.

Here is a list of my concerns/questions:

  • My main concern is the fact that the number of particles remains small: at the end some results (position space density in the ground state) are shown for the 1+6 problem, for which discrepancy with MPS is already larger. It is not clear how well the method can reproduce the energy spectrum (for 6 and for larger values of $N_{\uparrow}$). It is said: “we focus on the Fermi polaron problem”. The Fermi polaron problem is a single distinguishable impurity which interacts with a non-interacting Fermi sea. One or two or six particles is arguably a Fermi sea. The authors should address the question of larger values of $N_{\uparrow}$ (if not, I would not call it the Fermi polaron problem).
  • I guess the title is inspired by the title of Ref. 47. Here, however, a Gram-Schmidt procedure provides the basis in which the Hamiltonian is then numerically diagonalized. I therefore don’t like the term “interpolatory ansatz”, which suggests a simple form of the wave function with some variational parameters like in Ref. 47. I would urge the authors to choose a more suitable title, which also makes it clear that the method is quite different from the one presented in Ref. 47.
  • Many important references are missing and some references seem less relevant to me. For example: although Ref. [2] is indeed a seminal paper on the superfluid to Mott insulator transition in a 3D optical lattice, the discussion is about 1D (I could add many other breakthrough papers if we include 3D). No mention of other important experimental papers in 1D, such as e.g. Liao et al, Nature 467, 567 (2010) or Yang et al, Phys. Rev. Lett. 119, 165701 (2017). Many references about many-body techniques for the Fermi polaron problem are missing or to experiments (e.g. Nature 485, 615 (2002)). These are just a few examples. I would urge the authors to improve/extend their bibliography.
  • Is there a particular reason the discussion is limited to the repulsive case? Does the method have a problem with handling the attractive case? I think this should be addressed in the paper.
  • The first sentence of the abstract reads: “we explore a new variational principle for…”. This sounds very strange since the variational principle is what it is. You are not exploring a new one. Please rephrase this.
  • In section 5.1.1 the authors write “it is important to exclude such linearly dependent states to avoid singular behavior in the Gram-Schmidt orthogonalization process”. For the 1+1 problem, this can easily be done. Does this cause any problems for the general case of N_{\uparrow} particles?
  • Please include which units are being used in the plots (for energy, length, etc.)
  • Figure 2 shows the energy of the lowest six states for the 1+1 system in a harmonic trap. I expected the energies for the even-parity states to be quasi-identical to the ones plotted in Figure 1 of Ref. 47. (It is stated in Ref. 47 that the error there is bounded by about 0.03). However, the energies do not seem to agree. Why? Even if it’s just a matter of units or an energy shift, the relative energies of the fifth and the sixth states seem very different. Can the authors explain where the difference with Ref. 47 comes from?
  • In section 5.1.3 the authors write “we do not claim to have that high numerical precision in neither our method nor in the numerical integral used for the analytical formula.” Can the authors provide an estimate of their numerical/extrapolation error? This allows the reader to know up to which point to trust figure 5.
  • I compared figure 7 to figures 2 and 3 of Ref. 47. I again see quantitative and qualitative differences. Can the authors explain these differences?
  • In the conclusion it is stated that “computing densities is however a significantly time consuming step”. Can the authors be more precise about how time-consuming it is and why?

Finally, I found some problems with notations: - Notations are not always consistent with having n states at zero interaction and m states at infinite interaction. (See, e.g., page 3: $0 \leq i \leq n$, which says there are $n+1$ non-interacting states. This happens a number of times throughout the paper. - On page 3, $\mathcal{M}_l$ is the set of points where $x_0$ is SMALLER than exactly $l$ of the $x_1, \ldots, x_N$. On pages 5 and 13 it is LARGER and GREATER, while on page 8, it is again SMALLER. - In a number of places the superscripts (i) or ($\mu$) are dropped for the quantum numbers $k$ or $q$, but this not done in a consistent way (for example, in eq. (9) and on page 5 right above equation (11)). I suggest to include these everywhere. - Section 3.3 starts with a mere repetition of what was said right before, but this time with different notation: what was $a^{(\mu)}_j$ before is now $\alpha_j$. Even more confusingly, the notation $\alpha_j$ is reused at the end of 3.4 to indicate the final basis states. Please stick to a single consistent notation. - At the end of section 3.5., n and m suddenly indicate some many-body states. I didn’t quite understand what these states exactly are because the notation was not clearly introduced.

  • validity: -
  • significance: -
  • originality: -
  • clarity: -
  • formatting: -
  • grammar: -

Author:  Erik Jonathan Lindgren  on 2020-04-18  [id 799]

(in reply to Report 1 on 2019-12-06)
Category:
answer to question

We thank the referees for carefully reading the manuscript and for the comments and questions. We have modified and significantly improved our work based on these comments. Below we reproduce the referee's questions and offer a point-by-point answer to them.

1.My main concern is the fact that the number of particles remains small: at the end some results (position space density in the ground state) are shown for the 1+6 problem, for which discrepancy with MPS is already larger. It is not clear how well the method can reproduce the energy spectrum (for 6 and for larger values of N. It is said: “we focus on the Fermi polaron problem”. The Fermi polaron problem is a single distinguishable impurity which interacts with a non-interacting Fermi sea. One or two or six particles is arguably a Fermi sea. The authors should address the question of larger values of N (if not, I would not call it the Fermi polaron problem).

To exemplify how our method works also for larger systems, we have added a figure for the density of a system with 11 particles. We find that our method holds for this example as well. However, to avoid the issue pointed out by the referee, we also shift our description of the setup from the "polaron" to the "impurity" problem.

Regarding remaining imprecisions in the comparison of the energy spectrum with different methods, it is important to keep in mind that it is difficult to achieve arbitrary accuracy with MPS. While it serves as a good benchmark for most purposes, it is still approximative and is only able to simulate a continuous system to a certain extent. For instance, to approach the continuum we must define a very small lattice spacing, which in turn maps into an increasing value for the interaction parameter U in the Hubbard model with a single impurity. For very large systems, MPS is thus less efficient in finding the precise ground state energies and densities. This will thus lead to some additional discrepancy in the comparison of the methods.

2.I guess the title is inspired by the title of Ref. 47. Here, however, a Gram-Schmidt procedure provides the basis in which the Hamiltonian is then numerically diagonalized. I therefore don’t like the term “interpolatory ansatz”, which suggests a simple form of the wave function with some variational parameters like in Ref. 47. I would urge the authors to choose a more suitable title, which also makes it clear that the method is quite different from the one presented in Ref. 47.

The method uses states at zero interaction and infinite interaction, and then, by diagonalizing the Hamiltonian, finds a suitable combination that "interpolates" to finite values of g. While we certainly appreciate the referee's input about this and we have made some changes to the title, we believe that interpolatory is still a suitable word to describe the method, otherwise the title will be too long.

3.Many important references are missing and some references seem less relevant to me. For example: although Ref. [2] is indeed a seminal paper on the superfluid to Mott insulator transition in a 3D optical lattice, the discussion is about 1D (I could add many other breakthrough papers if we include 3D). No mention of other important experimental papers in 1D, such as e.g. Liao et al, Nature 467, 567 (2010) or Yang et al, Phys. Rev. Lett. 119, 165701 (2017). Many references about many-body techniques for the Fermi polaron problem are missing or to experiments (e.g. Nature 485, 615 (2002)). These are just a few examples. I would urge the authors to improve/extend their bibliography.

We thank the referee for pointing out more relevant references, and our references have been updated. We added not only the references suggested by the referee, but also a number of studies concerning the Fermi polaron problem (see refs. 27-36).

4.Is there a particular reason the discussion is limited to the repulsive case? Does the method have a problem with handling the attractive case? I think this should be addressed in the paper.

It is possible that (an extension of) the method would work also for attractive interactions (which would built on what was done in Ref 47), but since there are states that diverge to negative energy (and thus do not approach any finite state at infinite interaction), it would be necessary to add additional states to the basis to accurately describe these. While this is for sure an interesting direction to explore, we consider it out of scope for this paper and will leave it for future work. We have added a more detailed comment in the conclusions about this.

5.The first sentence of the abstract reads: “we explore a new variational principle for…”. This sounds very strange since the variational principle is what it is. You are not exploring a new one. Please rephrase this.

We have rephrased this sentence to not mention the "variational principle".

6.In section 5.1.1 the authors write “it is important to exclude such linearly dependent states to avoid singular behavior in the Gram-Schmidt orthogonalization process”. For the 1+1 problem, this can easily be done. Does this cause any problems for the general case of N_{\uparrow} particles?

This is equally trivial for any particle number, namely one just has to remove the totally antisymmetric states (these are also eigenstates to the full Hamiltonian and can be safely removed since they will anyway be orthogonal to any non-trivial eigenstates). We have made this more clear in the manuscript (making the comment apply more generally and moved it to section 3.4 where it is not specific to any particular particle number).

7.Please include which units are being used in the plots (for energy, length, etc.)

We have reinstated correct units in the figures, but still work in natural units throughout the paper.

8.Figure 2 shows the energy of the lowest six states for the 1+1 system in a harmonic trap. I expected the energies for the even-parity states to be quasi-identical to the ones plotted in Figure 1 of Ref. 47. (It is stated in Ref. 47 that the error there is bounded by about 0.03). However, the energies do not seem to agree. Why? Even if it’s just a matter of units or an energy shift, the relative energies of the fifth and the sixth states seem very different. Can the authors explain where the difference with Ref. 47 comes from?

For the harmonic oscillator, it is possible to factor out the center of mass movement (for two particles this would be change of basis to the center of mass coordinate and a relative coordinate between the two particles. For more particles this is done by going to Jacobi coordinates). This has been done in Ref 47, while we show the energy for the whole system including the center of mass excitations, and thus the energy spectra will be very different. The center of mass motion is independent of the interaction and thus adds standard linear harmonic oscillator energies.

To compare Figure 2 in our manuscript with Figure 1 in Ref 47, first note that the odd parity states (all states not affected by the interaction) have been removed. Now, the ground state in Figure 2 is the same as the ground state in Figure 1, but with the 0.5\hbar\omega ground state energy of the center of mass added. The third state in Figure 2 is then the ground state in Figure 1 but with the 1.5\hbar\omega of the first excited harmonic oscillator state of the center of mass added. The fifth and sixth states in Figure 2 are then the second state in Figure 1 (plus 0.5\hbar\omega from the ground state of the center of mass) and the ground state in Figure 1 plus an additional 2.5\hbar\omega from the center of mass in the second excited state.

We have added a comment about this in the manuscript.

9.In section 5.1.3 the authors write “we do not claim to have that high numerical precision in neither our method nor in the numerical integral used for the analytical formula.” Can the authors provide an estimate of their numerical/extrapolation error? This allows the reader to know up to which point to trust figure 5.

We managed to find the culprit for this discrepancy and increased the accuracy by improving the accuracy in one of the numerical integrals, now the figure does not have this problem anymore and will be updated for the resubmission (and the corresponding comment will be removed)

10.I compared figure 7 to figures 2 and 3 of Ref. 47. I again see quantitative and qualitative differences. Can the authors explain these differences?

Again, we plot energies for the whole system while Ref. 47 does not include the center of mass excitations. We have added a comment about this in the manuscript.

11.In the conclusion it is stated that “computing densities is however a significantly time consuming step”. Can the authors be more precise about how time-consuming it is and why?

It is more computationally heavy than calculating energies, since it requires calculating several integrals numerically for each position space coordinate for which we want to evaluate the density. For the majority density, we have to evaluate a double integral which further increases the computation time. These calculations scale directly with the number of position space coordinates on which we want to evaluate the densities, and while for the minority density each position space coordinate can be evaluated in parallel, this is not as easy for the majority density. It is possible that similar improvements can be made for the majority density and density matrices but we will leave further optimizations and improvements of these numerical techniques to future work. We have changed the comment in the manuscript to reflect these points.

We thank the referee for valid points about the notation, and all these problems have been fixed for the resubmission.

---

## Round 2 · Referee Report · Anonymous · 2020-5-2

Report

The authors have addressed my earlier comments and the revised presentation is much clearer. I recommend publication.

---

## Round 2 · Author Response

Resubmission with changes recommended by referee

---

## Round 2 · List of Changes

1.Title and abstract changed slightly
2.Added results for eleven particles, including new figure
3.Added more references
4.Added units to figures
5.Added comments on attractive interactions in conclusions
6.Improved notation in section 3.5
7.Added more comments in beginning of section 5
8.Figure 5 has been improved, which previously had problems for large number of basis states
9.Comment added in end of Section 3.4

You are currently on this page

Resubmission scipost_201908_00010v2 on 18 April 2020

---

## Editorial Decision

published